# Single-cell analysis of gastric signet ring cell carcinoma reveals cytological and immune microenvironment features

Weizhu Zhao[1,2,3,7], Yanfei Jia[4,7], Guangyu Sun[3], Haiying Yang[5], Luguang Liu[6], Xianlin Qu[6], Jishuang Ding[6], Hang Yu[6], Botao Xu[6], Siwei Zhao[6], Ligang Xing[1,2] ✉ & Jie Chai[6] ✉

Gastric signet ring cell carcinoma (GSRC) is a special subtype of gastric cancer (GC) associated with poor prognosis, but an in-depth and systematic study of GSRC is lacking. Here, we perform single-cell RNA sequencing to assess GC samples. We identify signet ring cell carcinoma (SRCC) cells. Microseminoprotein-beta (*MSMB*) can be used as a marker gene to guide the identification of moderately/poorly differentiated adenocarcinoma and signet ring cell carcinoma (SRCC). The upregulated differentially expressed genes in SRCC cells are mainly enriched in abnormally activated cancer-related signalling pathways and immune response signalling pathways. SRCC cells are also significantly enriched in mitogen-activated protein kinase and oestrogen signalling pathways, which can interact and promote each other in a positive feedback loop. SRCC cells are shown to have lower cell adhesion and higher immune evasion capabilities as well as an immunosuppressive microenvironment, which may be closely associated with the relatively poor prognosis of GSRC. In summary, GSRC exhibits unique cytological characteristics and a unique immune microenvironment, which may be advantageous for accurate diagnosis and treatment.

Gastric cancer (GC) is a common cancer worldwide that ranks fifth in incidence and fourth in mortality[1]. Fewer early-stage GC patients are diagnosed in China than in other countries, and late-stage GC patients account for the highest proportion of patients with large tumours at the time of diagnosis[2]. Despite advances in early-stage diagnosis and treatment, the current understanding of the biology of GC is still preliminary, and the 5-year survival rate of patients with advanced GC remains below 5%[3].

Gastric adenocarcinoma (GA) is the most common subtype of GC, accounting for 95% of all GC cases. GA is a highly heterogeneous disease[4,5] and is classified into different histological subtypes based on Lauren's classification system, including intestinal-type, diffuse-type and mixed-type[6]. Gastric signet ring cell carcinoma (GSRC) is classified as diffuse-type GA, which develops from poorly cohesive cells without gland formation[7]. GSRC is histologically diagnosed based on microscopic characteristics, specifically, the presence of signet ring cells in

[1]Department of Radiation Oncology, Shandong University Cancer Center, Jinan, Shandong, China. [2]Department of Radialogy Oncology, Shandong Cancer Hospital and Institute, Shandong First Medical University and Shandong Academy of Medical Sciences, Jinan, Shandong, China. [3]Department of Oncology, Binzhou People's Hospital Affiliated to Shandong First Medical University, Binzhou, Shandong, China. [4]Research Center of Basic Medicine, Jinan Central Hospital, Shandong First Medical University, Jinan, Shandong, China. [5]Department of Cardiology, Binzhou People's Hospital Affiliated to Shandong First Medical University, Binzhou, Shandong, China. [6]Department of Gastroenterological Surgery, Shandong Cancer Hospital and Institute, Shandong First Medical University and Shandong Academy of Medical Sciences, Jinan, Shandong, China. [7]These authors contributed equally: Weizhu Zhao, Yanfei Jia. ✉e-mail: xinglg@medmail.com.cn; jchai@sdfmu.edu.cn

over 50% of cancer cells, according to the guidelines of the World Health Organization (WHO). Cancer cells secrete a large amount of mucus, but most of it is not discharged from the cell. Rather, the large amount of mucus in the cytoplasm squeezes the nucleus to one side, making the cancer cell look like a signet ring, hence the name signet ring cell carcinoma (SRCC). GSRC has the characteristics of low differentiation, high malignancy, strong invasiveness, higher risk of metastasis, poor response to radiotherapy and chemotherapy, and its incidence is increasing yearly, especially in women[8–11].

The molecular mechanisms involved in the occurrence and heterogeneity of GSRC have been described using exome and transcriptome sequencing, and changes in several driving factors have been recognized[12–14], However, these sequencing data reflect the average expression level of the gene in bulk cells.They mask the molecular characteristics of different cell subgroups within the tissue and cannot truly reflect the specificity and heterogeneity of the gastric printed cells. In recent years, single-cell RNA sequencing (scRNA-seq) has been used to directly analyse gene expression and intracellular population heterogeneity at the single-cell level, define dynamic transformations in cell type and cell state, and recognize new cell subtypes, thereby improving the understanding of transcription kinetics and gene regulation in rare cells[15–17]. The method can also be used to analyse biological tissues and has been widely applied in tumour microenvironment analysis, including tumorigenesis and immune tolerance contexts[18]. Furthermore, several studies have profiled the single-cell landscape of immune cell heterogeneity and tumour cell heterogeneity in GC[19,20]. However, a detailed understanding of the cytological and immune microenvironment in GSRC at the single-cell resolution level remains elusive.

In this work, we analyse para-cancerous and GA tissues by scRNA-seq. We compare the differences between moderately/poorly differentiated adenocarcinoma(M/PDA) and GSRC in cytology and immune microenvironment. Our data provide an in-depth understanding of GSRC and may provide a resource for the accurate diagnosis and treatment of GSRC.

## Results

### scRNA-seq overview and identification of major cell types in para-cancerous and GA tissues

From December 2019 to December 2021, 13 patients with GA who underwent radical gastrectomy at Shandong Cancer Institute and Jinan Central Hospital (Jinan, China) were enrolled. No enrolled patient received any other therapies before surgery, such as radiotherapy, chemotherapy, and immunotherapy, or had other tumours. All patients were negative for a family history of tumours. The patients were divided into 4 groups based mainly on the degree of differentiation and the signet ring cell content, and no gender-based analysis was performed. Additional patient data are presented in Supplementary Table 1. scRNA-seq was used to analyse the samples of 13 GA patients. After filtering out low-quality cells, removing doublet reads, and correcting for batch effects, the transcriptomes of a total of 149,782 single cells were analysed (Fig. 1a). In the discovery cohort, 32,456 single cells were obtained from para-cancerous tissues from 5 patients, and 117,326 single cells were obtained from cancer tissues from 13 patients. The validation cohort was used to confirm the differences found in the discovery cohort (Fig. 1b). Using CellRanger, the official analysis software from 10× Genomics, the cells were divided into 26 clusters based on principal component analysis (PCA) and cluster analysis (Fig. 1c). Seurat software was utilized to analyse the gene expression differences among the cell clusters and to screen the genes that were upregulated in different cells (Fig. 1d). The distribution of 8 different cell clusters was finally determined based on unbiased cell type recognition (Fig. 1e). The cell clusters were named according to specific marker genes: B cells (expressing CD79B, CD79A, and MS4A1), endothelial cells (expressing VWF, CDH5, and PECAM),

epithelial cells (expressing CDH1, KRT8, and EPCAM), fibroblast cells (expressing PDGFRB, COL1O2, and DCN), mast cells (expressing SLC18A2, FCER1A, TPSB2, and KIT), myoid cells (expressing FCGR2A, CD163, and MRC1), smooth muscle cells (expressing TAGLN, RGS5, and ACTA2) and T cells (expressing TRBC2, CD2, and CD3E)[19]. To further understand the cell clustering, in-depth analysis was carried out from the aspects of cell source, grouping and cell proportion (Fig. 1f–h).

### Identification of malignant and nonmalignant epithelial cells

Cell clusters 5, 7, 15, and 19 were defined as epithelial cell clusters. Through cluster analysis of the epithelial cells, 20 subclusters (Fig. 2a) were obtained. According to marker genes cancer cells (expressing CLDN4, REG4, TTF3, and CEACAM6), mucous cells (expressing TFF1, MUC5AC, TFF2, and MUC6), chief cells (expressing PGA3 and PGA4), parietal cells (expressing ATP4A and ATP4B), and endocrine cells (expressing CHGA and CHGB)[19] (Fig. 2b), the subclusters were redivided into 5 subclusters. Subclusters 6, 11, 12, and 13 were identified as M/PDA cells, subclusters 14 and 16 were identified as chief cells, subclusters 18 and 19 were identified as parietal cells, subcluster 9 was identified as endocrine cells, and other subclusters were identified as mucous cells (Fig. 2c, d).

The proportion of mucous cells increased significantly in poorly differentiated adenocarcinoma with signet ring cell carcinoma (PDSRCC) and GSRC (Fig. 2e). According to the abovementioned findings, immunofluorescence analysis of mucous cell markers was conducted, and it was revealed that these markers were also expressed in SRCC cells (Fig. 2f). SRCC cells contained a great deal of mucous, thus, these cells may be identified as mucous cells. To further distinguish mucous and SRCC cells, tissue origin was considered. Subclusters 1, 4, 8, and 15 were independently derived from cancer tissues and were defined as SRCC cells, and subclusters 0, 2, 3, 5, 7, 10, and 17 were defined as mucous cells (Fig. 2g, h).

To verify the accuracy of clustering, the following steps were performed. First, to facilitate analysis, the chief cells, parietal cells, and endothelial cells were classified as nonmalignant epithelium, and their identification was relatively clear. The average expression levels of 80 cancer cell-elevated genes obtained from The Cancer Genome Atlas (TCGA) were used; the cancer-related score was compared between nonmalignant epithelium and M/PDA cells, and there was a significant difference between the two groups (Fig. 3a). The same results were also found between SRCC and mucous cells (Fig. 3b). Second, InferCNV is a tool used to analyse copy number variations (CNV) in tumour cells using scRNA-seq data. Based on the expression matrix of InferCNV, the CNV of M/PDA cells was significantly higher than that of nonmalignant epithelium and mucous cells, while the CNV of SRCC cells was lower than that of mucous cells (Fig. 3c, Supplementary Fig. 2). Moreover, to further verify the accuracy of the identification, the CNV scores of all subclusters were calculated based on InferCNV. The CNV scores of subclusters 6, 11, 12, and 13 were relatively high; thus, these subclusters were identified as M/PDA cells with characteristics typical of malignant tumours (Fig. 3d). The CNV score of SRCC cells was lower than that of M/PDA and mucous cells but higher than that of nonmalignant epithelial cells (t-test, $p < 0.001$) (Fig. 3e). Finally, volcano plot analysis revealed that REG4, S100A10, TSPAN8, OLMF4, and PHGR1 were upregulated in M/PDA cells compared with nonmalignant epithelial cells (Supplementary Fig. 3a). According to the results of the GSVA, M/PDA cells were mainly enriched in cancer-related signalling pathways contributing to tumour growth, proliferation, and metastasis, such as the epithelial-mesenchymal transition, E2F targets, PI3K/AKT/mTOR and KRAS signalling pathways (Supplementary Fig. 3b, Supplementary Data 1). GSVA was also performed to characterize different sources of mucous and SRCC cells.PDAucous cells, SRCC cells were enriched for signalling pathways such as the tumour necrosis factor-α (TNF-α), nuclear factor-κB (NF-κB), and transforming growth factor-β (TGF-β) signalling pathways, which are crucial for cancer development (Fig. 3f).

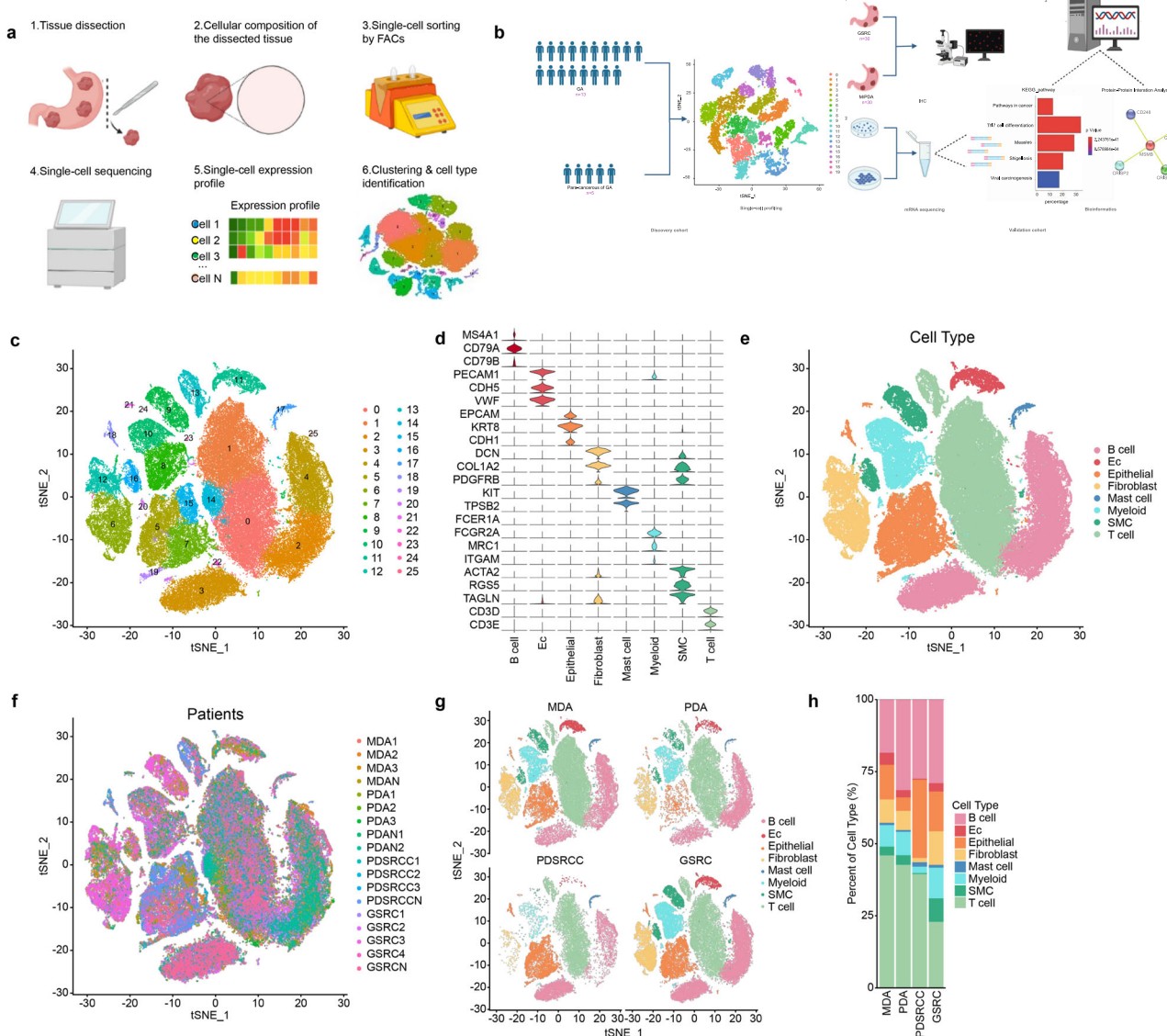

**Fig. 1 | scRNA-seq overview and identification of major cell types in para-cancerous and GA tissues. a** Schematic diagram of the scRNA-seq process. Designed with BioRender©. **b** Experimental design for scRNA-seq and the corresponding validation. In the discovery cohort, cancer tissues ($n = 13$) and para-cancerous tissues ($n = 5$) of 13 patients with GA who underwent radical gastrectomy were analysed. In the validation cohort, GSRC ($n = 30$) and M/PDA ($n = 30$) samples were used for IHC verification. The mRNA transcriptome was sequenced from cultured cells. Relevant databases were used for bioinformatics validation. Designed with BioRender©. **c** The t-distributed stochastic neighbour embedding (t-SNE) plot of the 26 main cell types identified from cancer tissues ($n = 13$) and para-cancerous tissues ($n = 5$); 32,456 single cells were obtained from para-cancerous tissues, and 117,326 single cells were obtained from cancer tissues. **d** Violin plot showing the expression levels of cell type marker genes in 8 cell types. **e** t-SNE plot of cells from cancer and para-cancerous tissues (coloured by cell type). **f** t-SNE plot of cells from cancer and para-cancerous tissues (coloured by sample origin). **g** t-SNE plots of cells from cancer tissues, which were grouped by the degree of differentiation and signet ring cell content. **h** Scale plot of cells from cancer tissues ($n = 13$), which were grouped by the degree of differentiation and signet ring cell content. Source data are provided as a Source Data file.

Collectively, the subclusters of epithelial cells were reidentified, and corresponding verification was carried out.

## scRNA-seq revealed the characteristics of SRCC
SRCC cells were divided into four subclusters (Fig. 4a). There was significant heterogeneity among SRCC cells according to heatmap analysis (Fig. 4b). However, in the four subclusters, some common highly expressed genes were identified, such as *CLDN18*, *PHLDA2*, *ATF3*, *HBEGF*, *SYTL2*, *PGC*, and *LYZ*, which might also be involved in the occurrence and development of GSRC.

According to the gene set enrichment analysis (GSEA) results, the genes upregulated in SRCC cells were mainly enriched in the immune response and included TNF-α signalling via the NF-kB

signalling pathway, the TGF-β signalling pathway, and the IL-18 signalling pathway. In addition, SRCC cells were enriched in cell proliferation-related signalling pathways, including the G2/M checkpoint, mitogen-activated protein kinase (MAPK), and TP63-targets signalling pathways (Fig. 4c). In addition, oestrogen response signalling and cholesterol homoeostasis signalling could be specifically enriched in SRCC cells, indicating that oestrogen receptor levels and cholesterol metabolism could be involved in cancer occurrence and development.

Importantly, *MSMB* was found to be a potential marker gene of GSRC. The expression level of *MSMB* was basically consistent with that in the subclusters of SRCC cells. *MSMB* exhibited essentially no/low expression in M/PDA cells (Fig. 4d, e). The scRNA-seq results

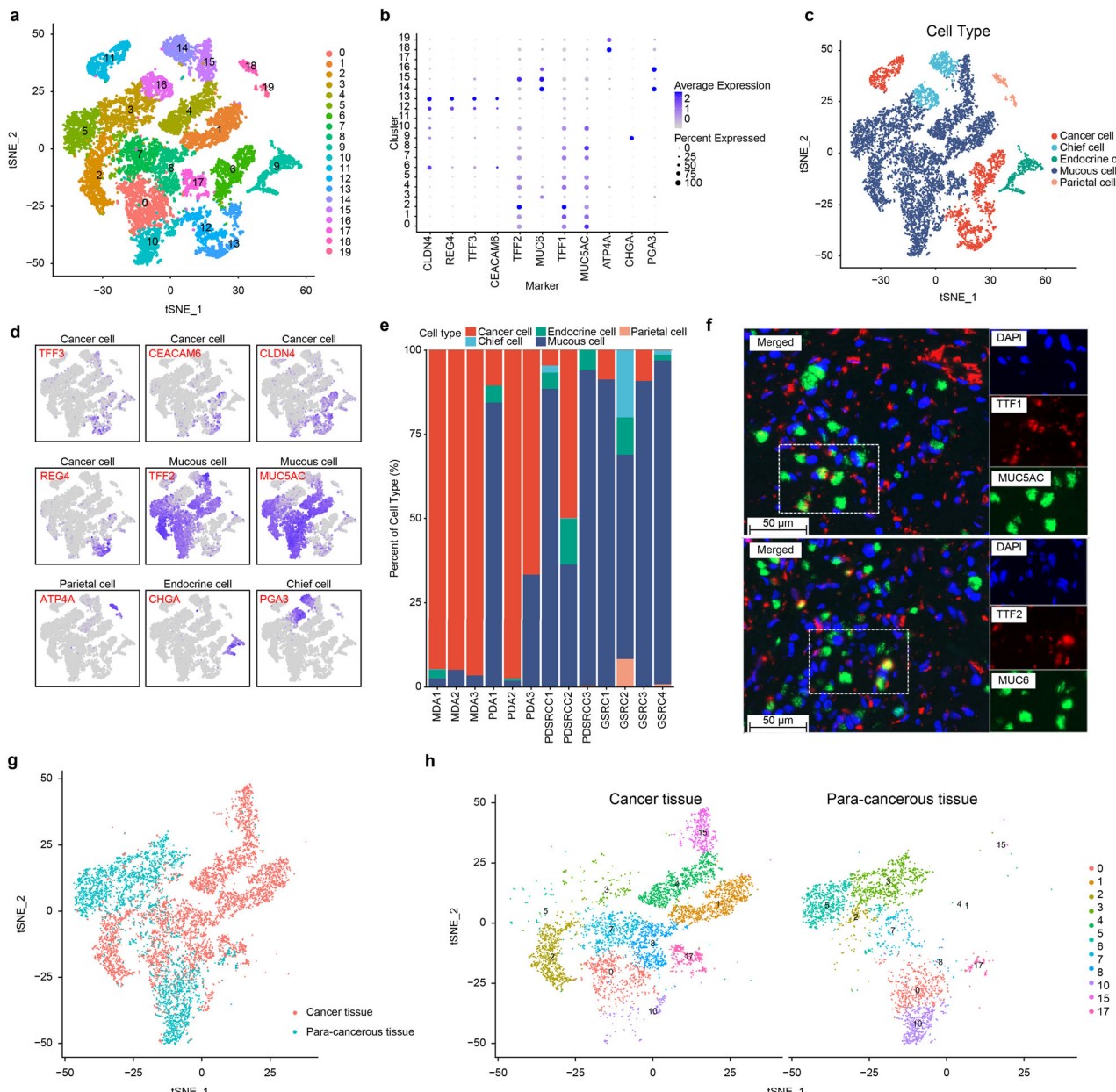

**Fig. 2 | Single-cell transcriptome plots of epithelial cells. a** t-SNE plot of 20 epithelial cells from cancer tissues ($n = 13$) and para-cancerous tissues ($n = 5$). **b** Dot plot of the expression levels of cell type marker genes in 5 cell types. **c** t-SNE plot of epithelial cells from cancer tissues ($n = 13$) and para-cancerous tissues ($n = 5$) (coloured by cell type). Subclusters 6, 11, 12, and 13 were identified as M/PDA cells, subclusters 14 and 16 were identified as chief cells, subclusters 18 and 19 were identified as parietal cells, subcluster 9 was identified as endocrine cells, and other subclusters were identified as mucous cells. **d** t-SNE plot of expression levels of marker genes for 5 cell types. **e** Scale plot of epithelial cells from cancer tissues

($n = 13$). **f** Immunofluorescence plot of characteristic genes of selected cell types ($n = 26$). The gastric antrum mucus cells were stained with DAPI (blue), *TTF1* (red) and *MUC5AC* (green). The gastric antrum basal gland mucus cells were stained with *MUC6* (green), *TTF2* (red), and DAPI (blue). Scale bar: 50 μm. **g** t-SNE plot of mucous cells from cancer tissues ($n = 13$) and para-cancerous tissues ($n = 5$) (coloured by tissue origin). **h** t-SNE plots of mucous cells of subclusters from cancer tissues ($n = 13$) and para-cancerous tissues ($n = 5$) (categorized by tissue origin and coloured by cell type). Source data are provided as a Source Data file.

showed that the expression level of *MSMB* was higher in SRCC cells than in mucous cells and M/PDA cells (Fig. 4f). Verification was carried out in the corresponding cell lines. The SRCC cell line NUGC4 exhibited a higher expression level of *MSMB* than the poorly differentiated MKN-45 adenocarcinoma cell line (Fig. 4g, h). Verification was conducted using IHC, and the results confirmed that the *MSMB* expression level in SRCC cells was higher than that in M/PDA cells but lower than that in the gastric foveal proliferation area in para-cancerous tissues (Fig. 4I). Taken together, these data supported the high *MSMB* expression level in SRCC cells.

## Similarities and differences between M/PDA and SRCC cells

Clinically, there were significant differences in biological behaviours between M/PDA and GSRC, and these differences were mainly manifested in the tumour parenchyma. To clarify these differences, DEGs were analysed in M/PDA and SRCC cells (Fig. 5a). Through the enrichment analysis of upregulated DEGs and signalling pathways, it was revealed thatPDA/PDA cells, the upregulated genes in SRCC cells were mainly enriched with abnormally active cancer-related signalling pathways (e.g., transcriptional misregulation in cancer, pathways in cancer and microRNAs in cancer) and

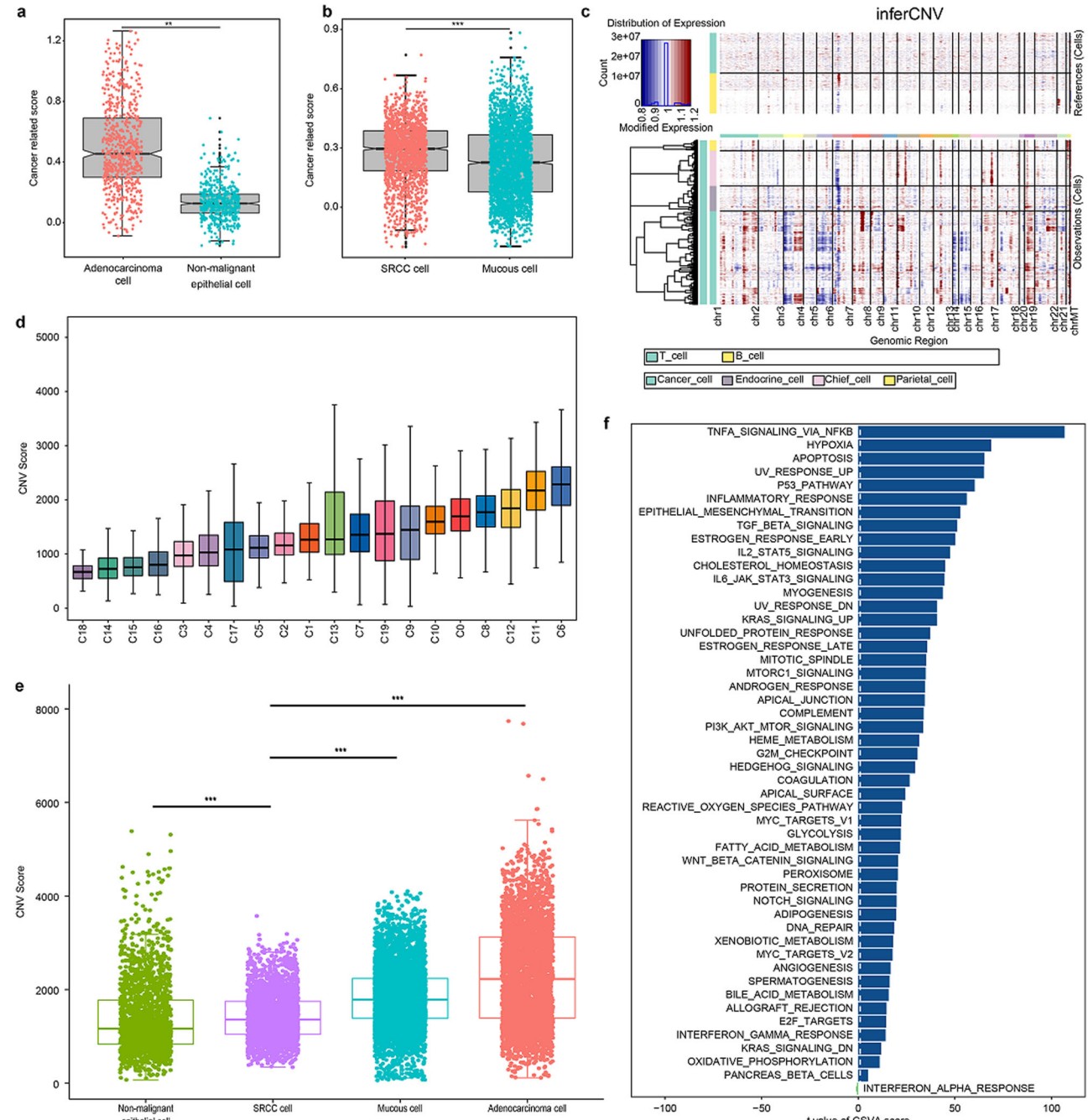

**Fig. 3 | Verification plots of nonmalignant epithelium and malignant epithelium after identification. a** Box plot showing distribution of cancer-related scores (average expression levels of cancer-related epithelial marker genes) for cells categorized as nonmalignant epithelial cells and M/PDA cells (*t*-test, $p < 2.2 \times 10^{-16}$). **b** Box plot showing distributions of cancer-related scores (average expression levels of cancer-related epithelial marker genes) for cells categorized as SRCC and mucous cells (*t*-test, $p < 2.2 \times 10^{-16}$). **c** Heatmap showing inferCNV for all subclusters of epithelial cells. Red: amplifications; blue: deletions. **d** Histogram showing CNV score plot of all subclusters of epithelial cells. **e** Box plot showing CNV score plot of epithelial cells by cell type. The CNV score of SRCC cells was lower than that of M/PDA and mucous cells, while higher than that of nonmalignant epithelial cells (*t*-

test, $p < 2.2 \times 10^{-16}$). **f** GSVA of SRCC and mucous cells. SRCC cells were mainly enriched in cancer-related signalling pathways such as the TNF-α signalling pathway, NF-κB signalling pathway, and TGF-β signalling pathway. GSVA data were plotted according to the *t* value of limma, and at value > 5 was considered significant. The statistical strategy were two-sided Student's *t*-test. In **a, b, d** and **e**, All specimens were participated in the analysis (*n* = 18), Data are presented as mean values ± SEM. In the box plot, the black dots represent outliers, the error bars represent SEM, the box midpoints represent means and the boxes represent interquartile positions. *p* values were calculated using the two-sided unpaired Student's *t*-test, P values < 0.05 were considered to indicate significance: **\**p* < 0.01; \**\**p* < 0.001. Source data are provided as a Source Data file.

were closely correlated with signalling pathways of immune escape (e.g., the TNF-α signalling pathway, TGF-β signalling pathway, and NF-kB signalling pathway) (Fig. 5b, c). The corresponding verification was carried out in cell culture, and the same results were

obtained (Fig. 5d, e). M/PDA cells had high proliferation and immune surveillance capabilities, while SRCC cells had low cell adhesion and high immune escape capabilities (*t*-test, *p* < 0.001). (Fig. 5f).

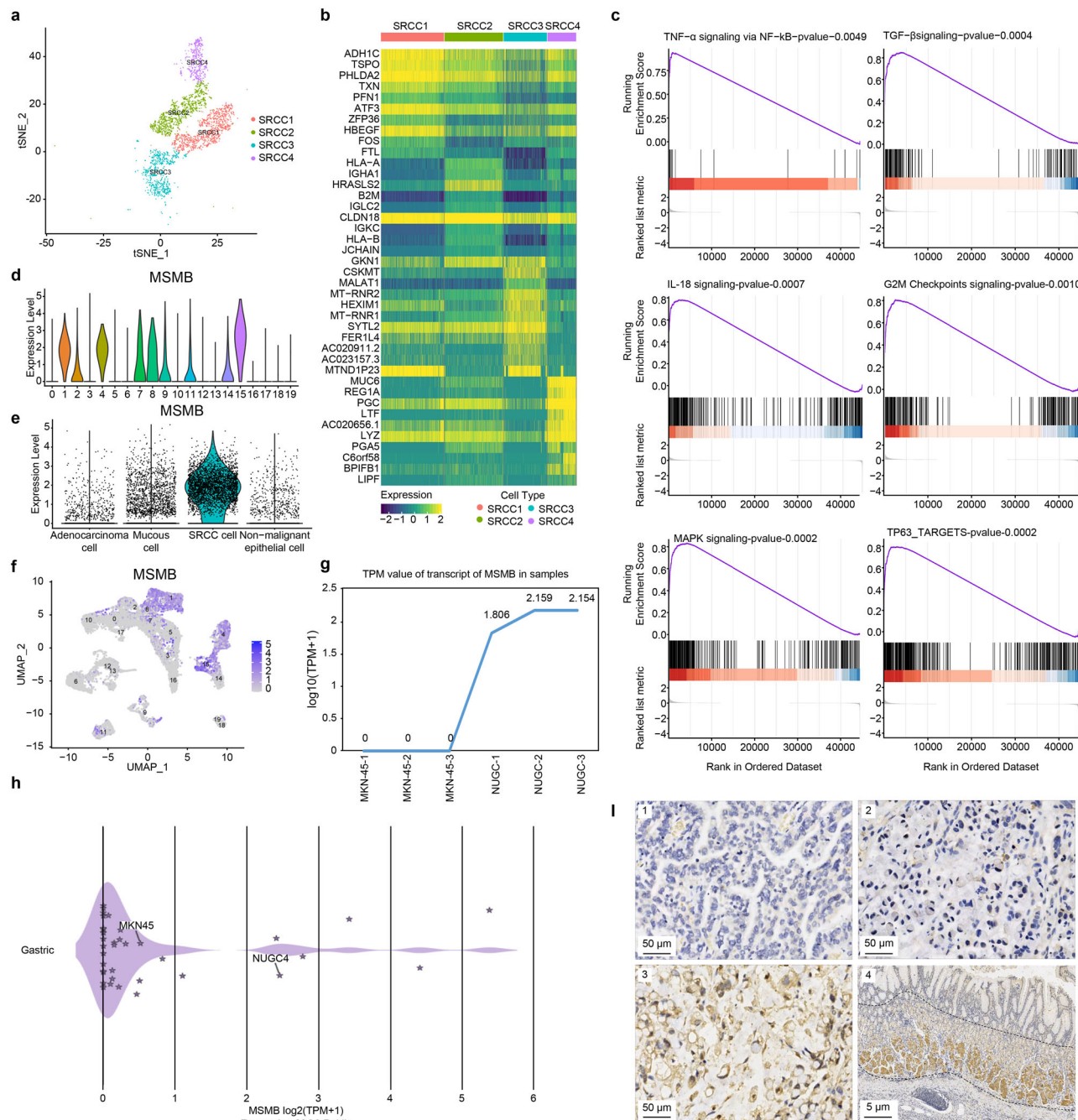

**Fig. 4 | Basic features of SRCC cells. a** t-SNE plot of 4 subclusters of SRCC cells. **b** Heatmap of 4 subclusters of SRCC cells. Some common highly expressed genes, such as *CLDN18*, *PHLDA2*, *ATF3*, *HBEGF*, *SYTL2*, *PGC*, and *LYZ* were identified and might also be involved in the occurrence and development of GSRC. **c** The results of GSEA showed that cancer-related signalling pathways were enriched in SRCC cells. The genes upregulated in SRCC cells were mainly enriched in signalling pathways associated with immune response, such as the TNF-α signalling pathway, NF-kB signalling pathway, and TGF-β signalling pathway, as well as cell proliferation-related pathways, including the G2/M checkpoint signalling pathway, MAPK signalling pathway, and TP63 signalling pathway. The statistical strategy were one-sided Student's *t*-test. **d** Violin plot of expression level of *MSMB* in subclusters. The expression level of *MSMB* was generally consistent with that in subclusters of 1, 4, 8,

and 15 of SRCC cells. *MSMB* is weakly or not at all expressed in M/PDA cells (subclusters of 0, 2, 3, 5, 7,10, and 17). **e** Violin plot of expression level of *MSMB* in epithelial cells(categorized by cell type). The expression level of *MSMB* was relatively higher in SRCC cells than mucous and M/PDA cells. **f** UMAP plot of the expression level of *MSMB* in subclusters. **g** TPM value of transcript of *MSMB* in MKN-45 and NUGC4 cells by cell culture. **h** TPM value of transcript of *MSMB* in cell lines by DepMap database. The transcriptional expression data of gastric cancer cell lines were downloaded from DepMap database (https://depmap.org/portal/). **i** IHC plots of the expression level of *MSMB* in cancer tissues and para-cancerous tissues (*n* = 69). 1: PDA; 2: PDSRCC; 3: GSRC; 4: para-cancerous tissues. 1, 2 and 3, Scale bar: 50 μm; 4, Scale bar: 5 μm. Source data are provided as a Source Data file.

## Infiltration characteristics of B cells in M/PDA and GSRC

B cells from clusters 3 and 4 were reclustered into 16 subclusters (Fig. 6b, Supplementary Fig. 4a–c) using the marker gene (Fig. 6a). Of these, subclusters 1, 2, 4, 6, 8 and 12 were defined as mucosa-associated

lymphoid tissue-derived B (MALT-B) cells (expressing immunoglobulin (Ig) and *JCHAIN*); subclusters 0, 3, 7, 14 and 15 were defined as follicular B cells (expressing *MS4A1*, *CD74* and *HLA-DRA*); subclusters 9, 10, 11 and 13 were defined as plasma cells (expressing Ig-related genes and

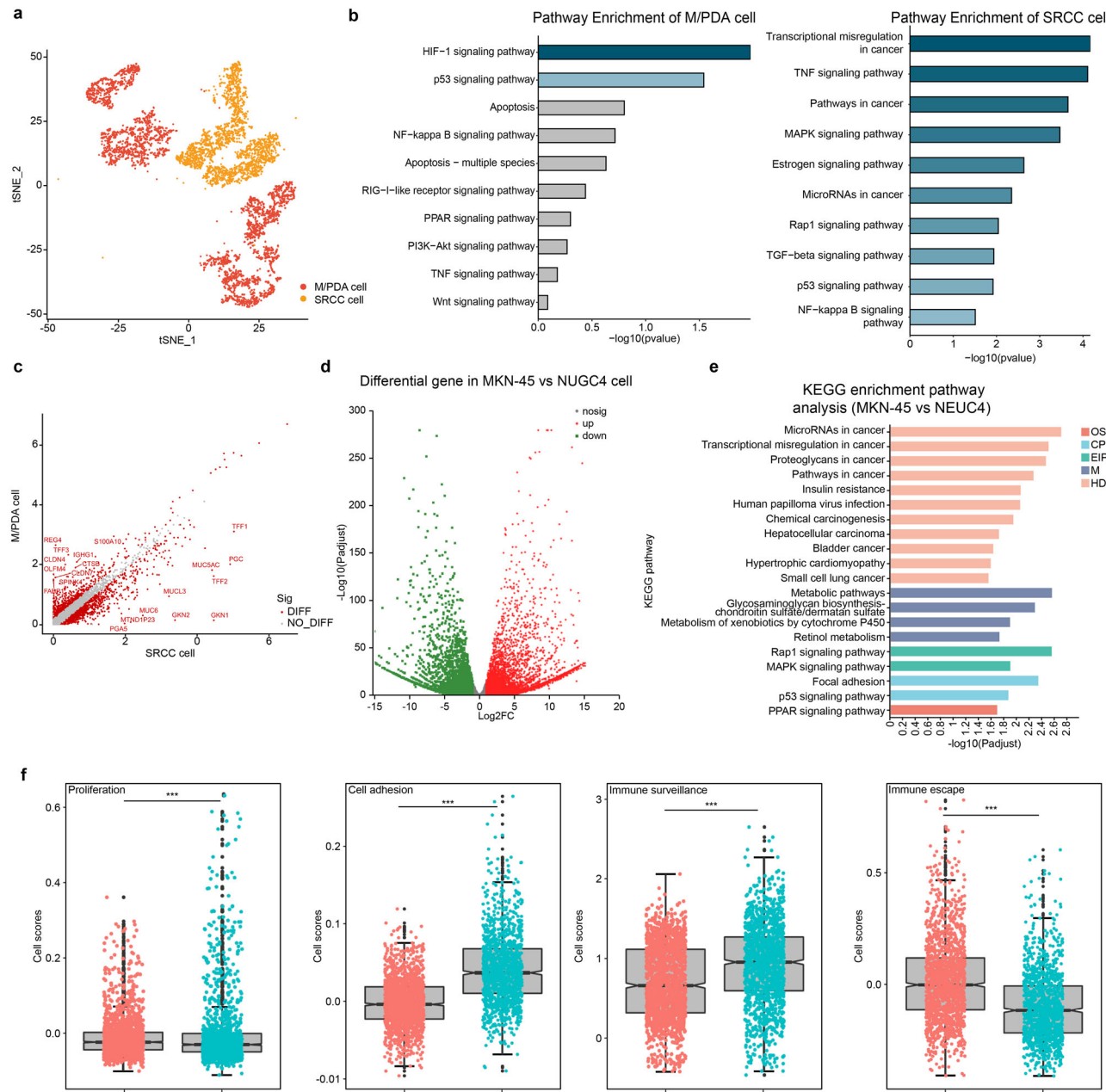

**Fig. 5 | Similarities and differences between M/PDA and SRCC cells. a** t-SNE plot of M/PDA and SRCC cells (coloured by cell type). **b** Pathway enrichment analysis plots of DEGs upregulated in M/PDA and SRCC cells. Compared with M/PDA cells, the DEGs upregulated in SRCC cells were mainly enriched in abnormally activated cancer-related signalling pathways (e.g., transcriptional misregulation in cancer, pathways in cancer and microRNAs in cancer) and were closely associated with signalling pathways of immune escape (e.g., TNF signalling pathway, TGF-β signalling pathway and NF-kB signalling pathway). **c** Heatmap plot of DEGs in M/PDA and SRCC cells. **d** Heatmap plot of DEGs in MKN-45 and NUGC4 cells. **e** KEGG pathway enrichment analysis of MKN-45 and NUGC4 cells among DEGs.

**f** Differences between M/PDA and SRCC cells in terms of cell proliferation, cell adhesion, immune surveillance, and immune escape. M/PDA cells had high proliferation and immune surveillance capabilities, while SRCC cells had low cell adhesion and high immune escape capabilities (t-test, $p < 0.001$). $n = 10$ (include 3 samples in MDA and PDA groups, respectively; and 4 samples in GSRC group). Data are presented as mean values ± SEM. In the box plot, the black dots represent outliers, the error bars represent SEM, the box midpoints represent means and the boxes represent inter-quartile positions. $p$ values were calculated using the two-sided unpaired Student's t-test, $p$ values < 0.05 were considered to indicate significance: ***$p < 0.001$. Source data are provided as a Source Data file.

*IGHG1*); and subcluster 5 was defined as memory B cells (expressing *HMGN2* and *H2AFZ*)[21,22].

After extracting the top 10 genes from each subcluster of B cells to form a heatmap, it was found that MALT-B cells mainly expressed IgA-related genes, while plasma cells mainly expressed IgG-related genes (Fig. 6c). MALT-B cells are similar to plasma cells in their functions and play an important role in the humoral immunity of B cells. Afterwards, enrichment analysis was performed on each subcluster, and it was found that MALT-B and plasma cells had similar functions, which were

mainly related to the immune response, B-cell activation, and participation in complement activation. However, follicular B cells had relatively weak immune responses (immune system process and regulation of B cells) (Fig. 6d).

Analysis of the samples revealed that the infiltration of MALT-B cells was the mainstay in MDA, PDA, and PDSRCC in para-cancerous tissues, while GSRC was mainly infiltrated by follicular B cells. The analysis of cancer tissue showed that MALT-B and memory B cells were the main infiltrating cells in MDA and PDA, while follicular B and

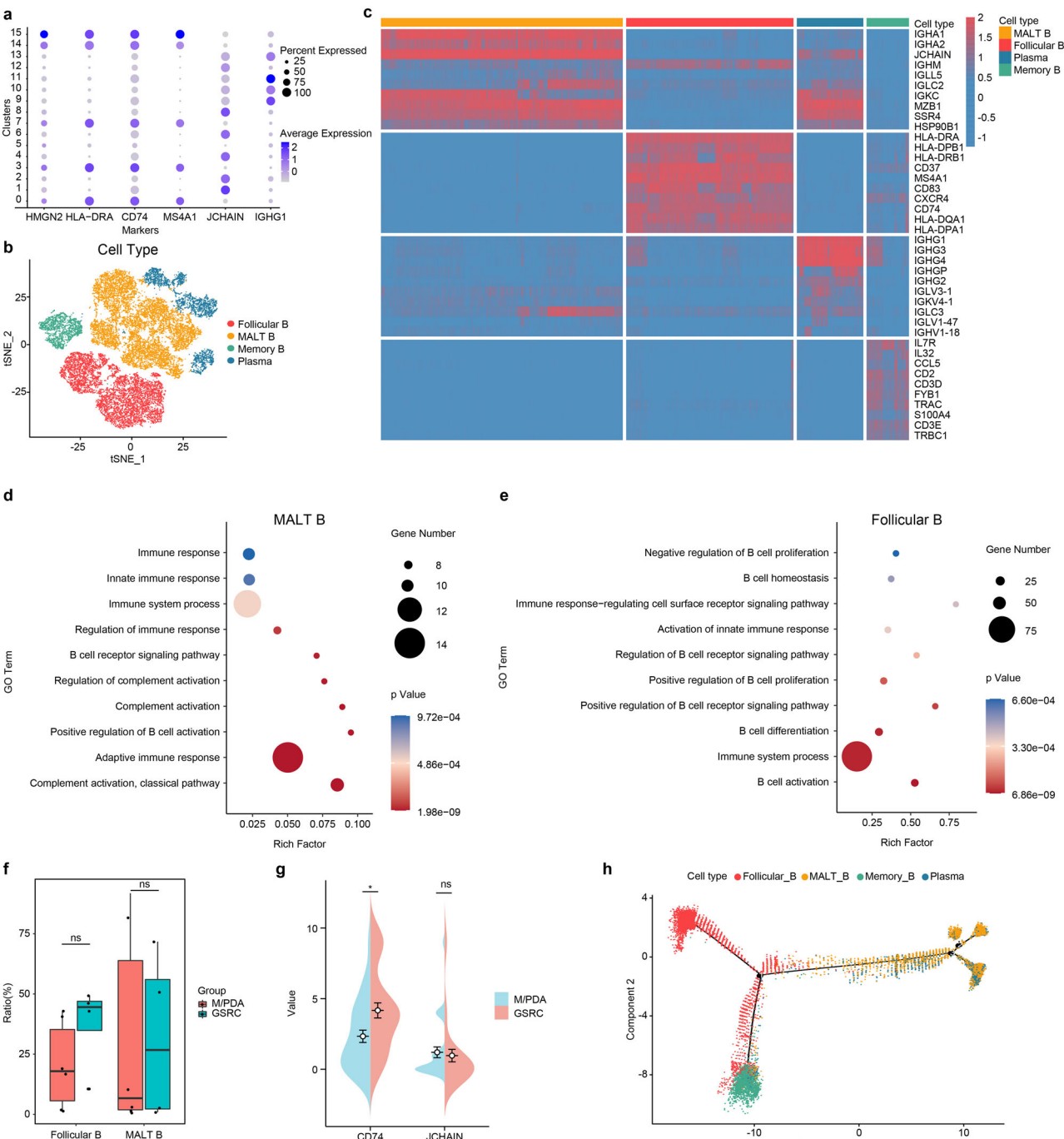

**Fig. 6 | Similarities and differences of tumour-infiltrating B cells between M/PDA and GSRC.** **a** Dot plot showing the expression levels of cell type marker genes in 4 cell types. **b** t-SNE plot showing subclusters of B cells (coloured by cell type). **c** Heatmap plot showing subclusters of B cells. **d** Functional signalling pathway enrichment plot of MALT-B cells. MALT-B cells were mainly related to immune response, B-cell activation, and participation in complement activation. **e** Functional signalling pathway enrichment plot of follicular B cells. Follicular B cells were related to immune system and regulation of B cells. **f** Box plot of tumour-infiltrating follicular B and MALT-B cells in M/PDA and GSRC. Compared with M/PDA, tumour-infiltrating follicular B cells in GSRC were higher, the difference was not statistically significant (t-test, *p* = 0.19); Similarly, there is no difference in tumour-infiltrating MALT-B cells between M/PDA and GSRC (t-test, *p* = 1). *n* = 10 (include 3 samples in MDA and PDA groups, respectively; and 4 samples in GSRC group). Data are presented as mean values ± SEM. In the box plot, the black dots

represent outliers, the error bars represent SEM, the box midpoints represent means and the boxes represent inter-quartile positions. **g** Pod plot of tumour-infiltrating follicular B and MALT-B cells in M/PDA and GSRC by IHC score (*n* = 30). Compared with that of M/PDA, the IHC score of tumour-infiltrating follicular B cells (expressing *CD74*) in GSRC was higher, and the difference was statistically significant (Wilcoxon rank-sum test, *p* = 0.012); The IHC score of tumour-infiltrating MALT-B cells (expressing *JCHAIN*) in GSRC was higher, but the difference was not statistically significant (Wilcoxon rank-sum test, *p* = 0.154). *p* values were calculated using Wilcoxon rank-sum test. **h** Pseudochronological analysis of subclusters of B cells. Starting from marker point 1, B cells were divided into two branches: one consisting of follicular B cells and memory B cells; the other consisting of MALT-B cells and plasma cells. In **d**, **e** and **f**, the statistical strategy were two-sided Student's t-test. *p* values < 0.05 were considered to indicate significance: \**p* < 0.05; ns no significance. Source data are provided as a Source Data file.

memory B cells were the main infiltrating cells in PDSRCC and GSRC. The increased proportion of SRCC was accompanied by a decreased proportion of MALT-B cells and a significant increase in the number of follicular B cells, which was highly consistent with para-cancerous tissues in GSRC (Supplementary Fig. 4d). Compared with M/PDA, tumour-infiltrating follicular B cells in GSRC were higher, the difference was not statistically significant (*t*-test, *p* = 0.19); Similarly, there was no difference in tumour-infiltrating MALT-B cells between M/PDA and GSRC (*t*-test, *p* = 1) (Fig. 6e). Compared with that of M/PDA, the IHC score of tumour-infiltrating follicular B cells (expressing *CD74*) in GSRC was higher, and the difference was statistically significant (Wilcoxon rank-sum test, p = 0.012); The IHC score of tumour-infiltrating MALT-B cells (expressing *JCHAIN*) in GSRC was higher, but the difference was not statistically significant (Wilcoxon rank-sum test, *p* = 0.154) (Fig. 6f, Supplementary Fig. 5).

To gain insight into the evolutionary relationship among B-cell subclusters, Monocle algorithm analysis was performed based on the definition of subclusters. The majority of cells from different subclusters were clustered according to the similarity of their gene expression, while different subclusters, which were clustered in pseudotime, illustrated a stepwise process, starting from marker point 1. B cells were divided into two branches: one consisting of follicular B cells and memory B cells; the other consisting of MALT-B cells and plasma cells (Fig. 6g).

In summary, compared with M/PDA, subclusters of B cells exhibited unique infiltration characteristics in GSRC, mainly follicular B cells. Moreover, follicular B cells had relatively weak immune responses; this could also be one of the reasons for the poor prognosis of GSRC.

## Infiltration characteristics of T cells in M/PDA and GSRC

T cells from clusters of 0, 1, 13, and 14 were reclustered into 14 independent subclusters (Fig. 7a). The subclusters were subdivided into 10 cell subclusters (Fig. 7c) using marker genes (Fig. 7b). Among them, the subcluster from cluster 0 was defined as CD4-Th17 cells (expressing *CD4*, *MAF*, *ICOS*, *CD40LG*, and *KLRB1*); the subcluster from cluster 1 was defined as CD4-Tn or central memory T (Tcm) cells (expressing *CD4*, *IL7R*, *TCF7*, *CCR7*, *SELL*, and *CD44*); subclusters from clusters 2, 3, 4, and 10 were defined as CD8-Teff cells (expressing *CD8A*, *NKG7*, *KLRD1*, and *GZMB*); the subcluster from cluster 5 was defined as CD8-Tex cells (expressing *CD8A*, *CTLA4*, *TIGIT*, *TNFRSF9*, and *PDCD1*); the subcluster from cluster 6 was defined as CD4-Treg cells (expressing *CD4* and *FOXP3*); the subcluster from cluster 7 was defined as naive T cells (expressing *CD3*); the subcluster from cluster 8 was defined as CD8-Tem cells (expressing *PRF1*, *GNLY*, *GZMH*, *FGFBP2*, *S1PR1*, and *S1PR5*); the subcluster from cluster 9 was defined as pro-T cells (expressing *MKI67*, *CD3D*, *CD8A*, and *CD4*); the subcluster from cluster 11 was defined as natural killer (NK) cells (expressing *CD3D*, *XCL1*, *TRDC*, and *XCL2*), and the subcluster from cluster 12 was defined as other cells[21,22].

Through the analysis of the samples, it was found that CD4-Th17, CD4-Tn or Tcm and CD8-Teff cells mainly infiltrated para-cancerous tissues and mainly exhibited immune functions. The infiltration rate of CD4-Treg cells was significantly higher in cancer tissues than in para-cancerous tissues, especially in GSRC. Meanwhile, GSRC had fewer tumour-infiltrating CD8-Teff cells than M/PDA (Fig. 7d). Compared with that in M/PDA, the proportion of tumour-infiltrating CD4-Treg cells in GSRC was higher, and the difference was not statistically significant (*t*-test, *p* = 0.25). The proportion of tumour-infiltrating CD8-Teff cells in GSRC was lower, the difference was not statistically significant (*t*-test, *p* = 0.25) (Fig. 6e). Compared with that in M/PDA, the IHC score of tumour-infiltrating CD4-Treg cells (expressing *FOXP3*) in GSRC was higher, and the difference was statistically significant (Wilcoxon rank-sum test, *p* = 0.0032); The IHC score of tumour-infiltrating CD8-Teff cells (expressing *KLRD1*) was lower in GSRC, and the difference was statistically significant (Wilcoxon rank-sum test, *p* = 0.0021) (Fig. 6f, Supplementary Fig. 5).

Collectively, the subclusters of T cells in GSRC exhibited unique infiltration characteristics, with increased infiltration of CD4-Treg cells and decreased infiltration of CD8-Teff cells, mainly manifesting as an immunosuppressive microenvironment.

## Discussion

GC is a common gastrointestinal malignancy and remains the fourth leading cause of cancer-related deaths worldwide[1]. Although recent epidemiological surveys have shown that the overall incidence of GC has decreased significantly, the incidence of diffuse GC is continuously increasing. This is especially true for GSRC due to its high heterogeneity, and it accounts for 3.4–45% of new GA cases[23–26]. Patients frequently exhibit advanced GSRC (stage III or IV) at the time of diagnosis, which is associated with poor prognosis. Therefore, this type of GC has particularly attracted oncologists' attention[27–30]. However, there is still a lack of in-depth understanding of the biological characteristics of GSRC, and further research is required to formulate targeted treatment strategies for GSRC.

In this work, relying on scRNA-seq, it is very important to accurately identify the SRCC cells. Starting from the unique characteristics of SRCC, that is, the large amount of mucus contained in the cells. The marker genes of gastric antrum basal gland mucus cells (*MUC6* and *TFF2*) and of gastric antrum mucus cells (*MUC5AC* and *TFF1*) were used for immunofluorescence examination[19], and it was confirmed that SRCC cells were identified in mucous cells. Further characterization, mainly depending on tissue origin, combined with comprehensive verification, such as cancer-related score, InferCNV, CNV score, and GSVA, identified SRCC cells. The degree of variation and CNV score of SRCC cells were lower than those of mucous and M/PDA cells. In general, tumour cells are prone to CNV mutations, and genes in the regions where CNV changes occur are always overexpressed or downregulated compared with normal cells. InferCNV estimated the single-cell CNV spectrum to distinguish between tumour cells and normal epithelial cells, which is irrelevant to the malignancy of tumours. The CNV score of cluster 15 was lower than that of the mucous cell clusters, which may have consequences on the total CNV score of clusters of SRCC cells. The role of tumour heterogeneity is noteworthy. According to the abovementioned results, the level of CNV does not discriminate benign and malignant GA cells, which is consistent with the results of another single-cell analysis, in which only 25.0% of static malignant cells exhibited high levels of CNV in GC[31]. In summary, InferCNV and CNV score provide favourable support for the regrouping of epithelial cells.

Furthermore, heatmap analysis revealed that there was also significant heterogeneity among SRCC subclusters. Similar to the case in other malignant tumours, tumour heterogeneity mainly has a noticeable influence on patient survival and prognosis[32], which may justify the poor prognosis of GSRC patients. Among the 4 subclusters, further analysis indicated that there were also some highly expressed genes among the subclusters, such as *CLDN18*, *PHLDA2*, *ATF3*, *HBEGF*, *SYTL2*, *PGC*, and *LYZ*. For instance, the importance of *CLDN18-ARHGAP26* was confirmed because of its frequent fusion in response to chemotherapy in GSRC[13]. High expression of *CLDN18.2* is considered to have potential value for targeted therapy of patients with advanced GSRC; the analysis of *CLDN18.2* expression and genetic abnormalities provides a new treatment option for advanced GSRC[33]. However, other genes have not been reported in studies on GSRC, and this area deserves further in-depth research.

In the process of studying GSRC, it was found that *MSMB* was highly expressed in SRCC cells but had no/low expression in M/PDA cells, consistent with previous studies[34,35]. In combination with data from relevant studies, our findings suggest that SRCC may originate from *MUC5AC*-/low *MUC6*- pre-pit cells in the proliferative zone of gastric glands[2]; SRCC of the gastric foveolar epithelium is positive for *MUC1*, *MUC5AC*, and *MUC6*, and SRCC, which is derived from the

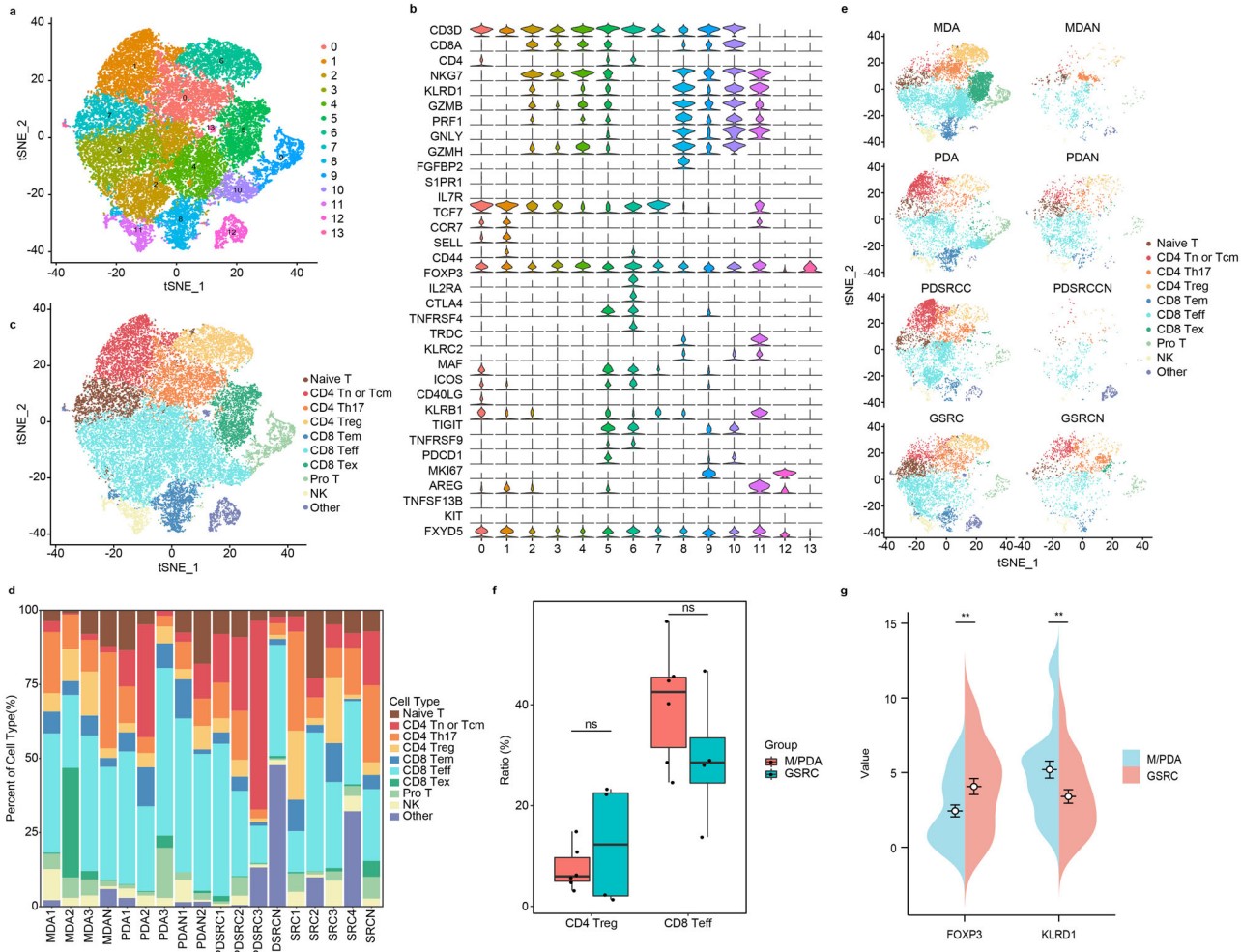

**Fig. 7 | Similarities and differences of tumour-infiltrating T cells between M/PDA and GSRC. a** t-SNE plot showing subclusters of T cells. **b** Dot plot showing the expression levels of cell type marker genes in 10 cell types. **c** t-SNE plot showing subclusters of T cells (coloured by cell type). **d** Scale plot of subclusters of T cells (*n* = 13). **e** t-SNE plots showing subclusters of T cells (coloured by group). **f** BOX plot of tumour-infiltrating CD4-Treg and CD8-Teff cells in M/PDA and GSRC. Compared with that in M/PDA, the proportion of tumour-infiltrating CD4-Treg cells in GSRC was higher, and the difference was not statistically significant (*t*-test, *p* = 0.25). The proportion of tumour-infiltrating CD8-Teff cells in GSRC was lower, the difference was not statistically significant (*t*-test, *p* = 0.25). *n* = 10 (include 3 samples in MDA and PDA groups, respectively; and 4 samples in GSRC group). Data are presented as mean values ± SEM. In the box plot, the black dots represent outliers, the error bars represent SEM, the box midpoints represent means and the boxes represent inter-quartile positions. *P* values were calculated using the two-side unpaired Student's *t*-test. **g** Differential plot of tumour-infiltrating CD4-Treg and CD8-Teff cells in M/PDA and GSRC by IHC score (*n* = 30). Compared with that of M/PDA, the IHC score of tumour-infiltrating CD4-Treg cells (expressing *FOXP3*) in GSRC was higher, and the difference was statistically significant (Wilcoxon rank-sum test, *p* = 0.0032); The IHC score of tumour-infiltrating CD8-Teff cells (expressing *KLRD1*) was lower in GSRC, and the difference was statistically significant (Wilcoxon rank-sum test, *p* = 0.0021). *p* values were calculated using the Wilcoxon rank-sum test. *p* values were calculated using the Wilcoxon rank-sum test. *p* values < 0.05 were considered to indicate significance: \*\**p* < 0.01; ns: no significance. Source data are provided as a Source Data file.

gastric foveolar epithelium, may originate from the proliferative region of the bottom of the gastric pit and gland neck. The IHC results confirmed the *MSMB* expression level in the gastric foveal proliferation area. The UMAP plot of *MSMB* also demonstrated that subclusters 7 and 8 were adjacent and were closely associated. Thus, a high *MSMB* expression level was detected in subcluster 7, while subclusters 0, 2, 3, 5, 10, and 17 were derived from mucous cells in other parts. SRCC may originate from subcluster 7. Therefore, *MSMB* was found to be a potential marker of GSRC and could be related to the differentiation and development of GSRC, and its specific biological function deserves further research.

The heterogeneity of GC is determined mainly by the tumour parenchyma and is closely correlated with its tumour microenvironment (TME). To clarify the significant differences in the biological behaviours between M/PDA and GSRC, the tumour parenchyma and TME were analysed.

Through the enrichment analysis of upregulated differentially expressed genes (DEGs), it was found that compared with M/PDA cells, SRCC cells were enriched for abnormally active cancer-related signalling pathways (transcriptional misregulation in cancer, pathways in cancer and microRNAs in cancer), which might be closely associated with the relatively poor prognosis and insensitivity to chemotherapy of GSRC[36–38]. The DEGs were also closely correlated with the signalling pathways of the immune response (TNF-α signalling pathway, TGF-β signalling pathway, and NF-kB signalling pathway), and the activity of the signalling pathways of the immune response in malignant tumours mainly indicates that the tumour has a high immune escape function[18]. The abovementioned pathway enrichment analysis results were highly consistent with the pathway enrichment analysis results of SRCC cell lines and adenocarcinoma cell lines in this study. Further analysis indicated that M/PDA cells had stronger proliferation and immune surveillance capabilities, while SRCC cells had weaker cell adhesion

and stronger immune evasion capabilities. GSRC cells appeared less sensitive to chemotherapy than M/PDA cells due to their low proliferation. Studies have demonstrated that changes in cell adhesion play an important role in the occurrence and development of GSRC. E-cadherin, encoded by the *CDH1* gene, is a cell–cell adhesion molecule that participates in the early formation of GSRC and plays a crucial role[39,40]. The frequency of cell adhesion-related gene mutations increased in GSRC[3]. PPI network analysis revealed 107 mutated genes related to cell adhesion, indicating that the cell adhesion pathway plays an important role in GSRC tumorigenesis[13]. However, it was revealed that the CDH1 expression level was essentially the same between M/PDA and SRCC cells, which could be related to the fact that GSRC was not detected in the early stage of formation. Compared with that in M/PDA cells, the intercellular adhesion function was decreased in SRCC cells; however, the intercellular adhesion function was decreased or destroyed, which is the key in causing cancer cells to break away from the parent tumour to initiate metastasis and may also be associated with higher risks of invasive growth and metastasis, especially for implanted metastases of GSRC.

The cell signalling pathways of M/PDA were enriched in the hypoxia-inducible factor 1 (*HIF-1*), *P53*, and PI3K-AKT signalling pathways, which is consistent with previously reported findings[22]. SRCC cells could also be significantly enriched in the MAPK signalling pathway, which is involved in tumour growth, proliferation, and metastasis[41,42]. This may be one of the intrinsic reasons for the high levels of invasion and metastasis of GSRC. In addition, it was found that the oestrogen signalling pathway could also be abnormally activated in SRCC cells, confirming that oestrogen could play an important role in the occurrence and development of GSRC[43,44]. Previous studies demonstrated that young female GA patients are prone to GSRC, and more than 80% of GSRC cells can secrete mucin and express oestrogen receptors, which are more likely to metastasize to the ovary, indicating that GSRC has a higher affinity for oestrogen and can promote tumour growth and invasion[45,46]. The oestrogen and MAPK signalling pathways can interact and promote each other through membrane-initiated steroid signalling[47–49]. Oestrogen signalling could also be activated by nuclear-initiated steroid signalling and can be regulated by HPS70, HPS90, and *FKBP5*[50–52]. In this study, heat shock protein family members were significantly enriched in the oestrogen signalling pathway. It was indicated that the nuclear-initiated signalling of oestrogen was one of the important pathways for its function in GSRC. Collectively, these findings show that blocking the MAPK and oestrogen signalling pathways is a potential method for the treatment of GSRC, and the associated specific internal mechanisms should be explored further.

Tumour tissues are composed of not only tumour cells themselves but also the cells in the TME, such as immune cells, fibroblasts, vascular endothelial cells, and stromal cells. Among these cells, tumour-infiltrating immune cells (TICs) play a vital role in tumorigenesis, development, and therapy[53,54]. Therefore, quantitative analysis of TICs in GC, especially comparative analysis of the immune microenvironment of M/PDA and GSRC, may clarify the characteristics of the mechanism of the immune response in the occurrence and development of GC, thereby laying a theoretical foundation for immunotherapy[55].

The subclusters of B cells in GSRC have a unique infiltrating feature, dominated by the infiltration of follicular B cells, whose main function is to activate immune system processes and regulate B cells[56]. Their performance in the immune response is relatively weak, indicating a relatively suppressed immune microenvironment, and the same results were obtained from functional cluster analysis and quasi-sequential analysis[40,57,58]. Follicular B cells account for approximately 25% of the total number of cells, which is indicative of the importance of B cells in antitumour immunity in GA and suggests one of the reasons for the poor prognosis of GSRC.

The infiltration rate of CD4-Treg cells in cancer tissues was significantly higher than that in para-cancerous tissues, especially in GSRC. Moreover, there were more infiltrating FOXP3+ CD4-Treg cells in GSRC than in M/PDA, and the infiltrating proportion of CD8-Teff cells in cancer tissues was lower than that in para-cancerous tissues, especially in GSRC. The current study revealed that Treg cells could inhibit the functions of effector T lymphocytes, and the positive expression of *FOXP3* was its main feature. The higher the positive rate of FOXP3+ CD4-Treg cells is, the faster the progression of GC and the lower the survival rate[59,60]. This may also be one of the reasons for the poor prognosis of GSRC.

Taken together, the findings here show that the subclusters of B and T cells in GSRC have unique infiltration characteristics. The infiltration of follicular B and CD4-Treg cells increased, and that of CD8-Teff cells decreased, in GSRC. Therefore, GSRC has an immunosuppressive microenvironment, which could be closely associated with the relatively poor prognosis and poor efficacy of immunotherapy. Regrettably, the underlying mechanisms need further study.

In conclusion, 32,456 single cells from para-cancerous tissues of 5 patients and 117,326 single cells from cancer tissues of 13 patients were analysed by scRNA-seq. We have identified SRCC cells. *MSMB* could be used as a marker to guide the identification of SRCC and M/PDA cells. The DEGs upregulated in SRCC were mainly enriched in abnormally activated cancer-related signalling pathways and signalling pathways of the immune response. SRCC could also be significantly enriched in the MAPK and oestrogen signalling pathways, which could interact and promote each other and continue to amplify each other's effects. SRCC cells exhibited lower cell adhesion and higher immune evasion capabilities, as well as an immunosuppressive microenvironment, which could be closely associated with the relatively poor prognosis of GSRC. One limitation of the present study is that it is difficult to collect sufficient samples and lack of analytical methods for multi-omics. Overall, compared with M/PDA cells, GSRC cells have unique cytological characteristics and a unique immune microenvironment, which may be advantageous for the accurate diagnosis and treatment of GSRC.

## Methods

### Ethics

The present study was approved by the Institutional Review Boards of Shandong Cancer Institute (Approval No. SDTHEC2020011006) and Jinan Central Hospital (Approval No. GZR2019-041-01). All specimens were obtained with written informed consent and were fully anonymized. Consent to publish relevant clinical information potentially identifying individuals (e.g., age, gender, histological grade, etc.) was obtained. We have consent to publish the information that identifies individuals. Participants were not granted any corresponding compensation, as our study exclusively utilized residual tissue samples (secondary use of surplus materials), effectively obviating the necessity for supplementary patient follow-up or communication. This research was conducted according to the principles of the Declaration of Helsinki.

### Human specimens

The gastroscopic biopsy pathology of enrolled patients at first diagnosis was GA; they had not received antitumour treatments, such as radiotherapy, chemotherapy, and immunotherapy before surgery, and did not have tumours in other organs. In the discovery cohort, 13 patients were prospectively enrolled for scRNA-seq, including 7 males and 6 females with median ages of 63 and 51 years old. The patients were grouped mainly based on the degree of differentiation and signet ring cell content, and no gender-based analysis was performed. In the validation cohort, 60 patients were retrospectively enrolled for immunohistochemistry (IHC).

### Tissue dissociation and preparation of single-cell suspensions

Human gastric tissue was obtained from patients and placed into a sterile RNase-free culture dish containing an appropriate amount of calcium-free and magnesium-free 1× phosphate-buffered saline (PBS) on ice. The tissue was then transferred into the culture dish, cut into 0.5 mm² pieces, and washed with 1× PBS, and blood residue and fatty layers were removed. Tissues were dissociated into single cells in dissociation solution. The overall cell viability was more than 85%, as confirmed by trypan blue exclusion. The single-cell suspensions were counted using the Countess II Automated Cell Counter, and the concentration was adjusted to 700–1200 cells/µl before single-cell analysis.

### 10X Genomics Chromium library and sequencing

Single-cell suspensions were loaded onto a 10X Genomics Chromium instrument to capture 8000 single cells according to the manufacturer's instructions for the 10X Genomics Chromium Single-Cell 3′ kit (V3). The following cDNA amplification and library construction steps were performed according to the standard protocol. Libraries were sequenced on an Illumina NovaSeq 6000 sequencing system (paired-end multiplexing run, 150 bp) by LC-Bio Technology Co., Ltd, (Hangzhou, China) at a minimum depth of 20,000 reads per cell.

CellRanger software (v7.0.0, 10X Genomics) was then used to analyse the sequencing data, and gene expression information was obtained for each cell. Cell Ranger (http://support.10xgenomics.com/single-cell/software/overview/welcome) used the STAR aligner (https://github.com/alexdobin/STAR) to perform splicing-aware alignment of reads to the genome. The Cell Ranger output was loaded onto Seurat (v4.1.1) software for dimensionality reduction, clustering, and analysis of scRNA-seq data. Overall, 149,782 cells passed the quality control threshold; all genes expressed in less than three cells were removed, the number of genes expressed per cell was between 500 and 5000, with a unique molecular identifier (UMI) count less than 500, and the rate of mitochondrial gene expression was <25%.

DoubletFinder (v2.0.3) was used to remove multiplet cells from sequencing data. To visualize data, the LogNormalize method of the "Normalization" function of Seurat software was utilized to calculate the expression levels of the genes. PCA was performed using the normalized expression levels, and the top 10 PCs were used to carry out clustering and t-distributed stochastic neighbour embedding (t-SNE) analysis. Due to the obvious batch effect among samples, Harmony (v0.1.0) was used for batch effect correction.

### Gene Ontology (GO) and Kyoto Encyclopedia of Genes and Genomes (KEGG) pathway enrichment analyses of differentially expressed genes (DEGs)

GO enrichment analysis provides all GO terms that are significantly enriched in DEGs compared with the genome background and filters the DEGs that correspond to biological functions. First, all related genes were mapped to GO terms in the Gene Ontology database (http://www.geneontology.org/), gene numbers were calculated for every term, and GO terms significantly enriched in DEGs compared with the genome background were defined by hypergeometric test.

Pathway-based analysis helps to further improve the understanding of the biological functions of genes. KEGG is the major public pathway database. Pathway enrichment analysis identified significantly enriched metabolic pathways or signal transduction pathways in DEGs compared with the whole genome background.

### Gene set enrichment analysis (GSEA)

Using the "clusterProfiler" R package, GSEA was performed on the GO and KEGG databases for RNA-seq data. GO and KEGG pathway enrichment analyses were carried out, followed by visualization using the "clusterProfiler" R package.

### Gene set variation analysis (GSVA)

GSVA is a nonparametric unsupervised analysis method that is mainly used to evaluate gene set enrichment results of microarray nuclear transcriptomes. It is typically utilized to indicate whether different metabolic pathways are enriched between different samples by converting the expression matrix of genes, mainly from the perspective of bioinformatics, to explain the causes of phenotypic differences.

### DepMap database

DepMap is a cancer cell line database that integrates and provides the existing cell line database. The transcriptional expression data of gastric cancer cell lines were downloaded from the DepMap database (https://depmap.org/portal/).

### InferCNV

InferCNV is a relatively efficient analytical tool for determining intracellular chromosomal copy number variation (CNV) changes. Tumour RNA-seq data were explored using InferCNV to analyse somatic large-scale CNVs, such as gains or losses of whole chromosomes and large segments of chromosomes. In general, tumour cells are prone to CNV mutations, and genes in the regions where CNV changes occur are always overexpressed or downregulated compared with normal cells.

### Cell culture

MKN-45 (CBP60488) and NUGC4(CBP60493) cells were purchased from Nanjing Kebai Biotechnology Co., Ltd (Nanjing, China). MKN-45 and NUGC4 cells were authenticated using short tandem repeat analysis. No mycoplasma contamination was detected. MKN-45 and NUGC4 cells cultured in Roswell Park Memorial Institute (RPMI)−1640 medium (HyClone, Logan, UT, USA) containing 10% foetal bovine serum (FBS; Hangzhou Sijiqing Co., Ltd., Hangzhou, China) at 37 °C in the presence of 5% $CO_2$.

### RNA extraction

Total RNA was extracted from the cells using TRIzol® reagent (RNA isolation from plant tissue) according to the manufacturer's instructions (Invitrogen, Carlsbad, CA, USA), and genomic DNA was removed using DNase I (TaKaRa, Shiga, Japan). Then, RNA quality was determined by the 2100 Bioanalyzer system (Agilent Technologies, Inc., Santa Clara, CA, USA) and quantified using the ND-2000 spectrophotometer (NanoDrop Technologies, Wilmington, DE, USA). Only high-quality RNA samples (optical density (OD) 260/280 = 1.8 - 2.2, OD260/230 ≥ 2.0, RNA integrity number (RIN) ≥ 6.5, 28S:18S ≥ 1.0, >1 µg) were used to construct the sequencing library.

### Library preparation and Illumina HiSeq X Ten/NovaSeq 6000 sequencing

An RNA-seq transcriptome library was constructed following TruSeqTM RNA sample preparation using a kit (Illumina Inc., San Diego, CA, USA) with 1 µg of total RNA. Briefly, messenger RNA (mRNA) was isolated according to the poly(A) selection method by oligo(dT) beads and then initially fragmented by fragmentation buffer. Second, double-stranded cDNA was synthesized using a SuperScript double-stranded cDNA synthesis kit (Invitrogen) with random hexamer primers (Illumina). Afterwards, the synthesized cDNA was subjected to end repair, phosphorylation, and 'A' base addition according to the Illumina library construction protocol (Illumina Inc.). Libraries were size-selected for cDNA target fragments of 300 bp on 2% Low Range Ultra Agarose, followed by polymerase chain reaction (PCR) amplification via Phusion DNA polymerase (NEB) for 15 PCR cycles. After quantification by TBS380, a paired-end RNA sequencing library was sequenced using the Illumina HiSeq X Ten/NovaSeq 6000 sequencer (2 × 150-bp read length). Data were analysed by the Majorbio cloud platform (www.majorbio.com).

## Immunohistochemistry (IHC)

IHC was performed with a HistoMouse SP Broad Spectrum DAB kit (Invitrogen) following the manufacturer's instructions. Briefly, samples were routinely deparaffinized, and antigen retrieval was conducted at high temperature and high pressure. Subsequently, the samples were incubated with primary antibody at 4 °C overnight. Then, 3,3′-diaminobenzidine (DAB) colour was developed, and each section was counterstained with haematoxylin for 30 s. Antibody was replaced with PBS as a blank control. The following antibodies were used to detect the proteins: anti-EpCAM (rabbit, 1:400, bs-0593R, Bioss, China), anti-TTF1 (rabbit, 1:1000, 66034-1-IG, 2C8F3, PROTEINTECH GROUP, USA), anti-TTF2 (rabbit, 1:100, 13722-1-AP, PROTEINTECH GROUP, USA), anti-MUC5AC (mouse,1:200, GB14112, l107, Servicebio, China), anti-MUC6 (mouse, 1:50, GB14113, l108, Servicebio, China), anti-JCHAIN (rabbit, 1:1000, bsm-60277R, J2H8, Bioss, China), anti-MSMB (rabbit, 1:200, bs-19185R, Bioss, China), anti-CD74 (mouse, 1:1000, GB121179, 3H10A4, Servicebio, China), anti-FOXP3 (rabbit, 1:400, GB11093, Servicebio, China), antiKLRD1 (rabbit, 1:100, DF6773, Affinity Biosciences, USA).The histopathological images of the cases were examined by two experienced pathologists. The results were judged by the semiquantitative integral method, and the score was comprehensively assessed according to the percentage of positive cells and the positive intensity. (1) Percentage of positive cells: ≤5% positive cells, 0 points; 6–25%, 1 point; 26–50%, 2 points; 51–75%, 3 points; and >75%, 4 points. (2) Positive intensity: colourless, 0 points; pale yellow, 1 point; yellow, 2 points; and brownish yellow; 3 points. The final score was calculated as follows: final score = positive ratio score × positive intensity score.

## Statistical analysis

Marker genes for transcriptional subpopulations in scRNA-seq profiles were identified using the FindAllMarkers function of Seurat software with a minimum log-fold change threshold of 0.25 and with $p$ values calculated by the Wilcoxon rank-sum test. $p$ values were computed using a hypergeometric test and adjusted for multiple hypothesis testing with the Benjamini–Hochberg procedure. The experimental data were statistically analysed using SPSS 24.0 software (IBM, Armonk, NY, USA). The $t$-test and Wilcoxon rank-sum test were utilized to compare mean values between two groups. The Bonferroni correction method was used to make pairwise comparisons when the difference was statistically significant; correlations were explored using Pearson's correlation analysis. $p < 0.05$ was considered statistically significant.

## Reporting summary

Further information on research design is available in the Nature Portfolio Reporting Summary linked to this article.

## Data availability

The raw scRNA-seq data used in this study are available in the Genome Sequence Archive (Genomics, Proteomics & Bioinformatics 2021) in the National Genomics Data Center (Nucleic Acids Res 2022), China National Center for Bioinformation/Beijing Institute of Genomics, Chinese Academy of Sciences database under accession code GSA-Human: HRA003647. The raw RNA-seq data generated in this study have been deposited in the Genome Sequence Archive (Genomics, Proteomics & Bioinformatics 2021) in the National Genomics Data Center (Nucleic Acids Res 2022), China National Center for Bioinformation/Beijing Institute of Genomics, Chinese Academy of Sciences database under accession code GSA-Human: HRA003650. The published data used for validation of the expression of *MSMB* in cell lines in this manuscript were retrieved from DepMap databases (https://depmap.org/portal/Interactive). The data are provided in the form of a single Excel file and placed in multiple label files in a compressed folder. This file or folder was also named "Source Data". The remaining data are also available in this manuscript. Source data are provided with this paper.

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

## Acknowledgements

This study was supported by the Major Innovation Project of Science and Technology of Shandong Province (No. 2019ZZZY011008, by J.C.), the Natural Science Foundation of Shandong Province (No. ZR2021MH108,

by J.C.), the Academic Promotion Program of Shandong First Medical University (No. 2019QL024, by Y.J.), and the Hospital-level project of Binzhou People's Hospital (No. XJ202202505, by W.Z.). We thank LC-Bio Technology Co., Ltd (Hangzhou, China) and Majorbio for their assistance in sequencing.

## Author contributions

L.X. and J.C. were the principal investigators who conceived and designed the study, obtained financial supports and approved the final version of the manuscript; W.Z. and Y.J. performed data acquisition and analysis and wrote and revised the manuscript.; G.S., L.L., X.Q. and J.D. collected clinical samples and performed immunohistochemistry; H.Y. and H.Yu performed data compilation and statistics; B.X. and S.Z. were responsible for cell culture. All the authors read and approved the final version of the manuscript.

## Competing interests

The authors declare no competing interests.
