## [Peer Review File · Nature Communications]

Single-cell analysis of gastric signet ring cell carcinoma reveals cytological and immune microenvironment featuresReviewers' Comments:

Reviewer #1:

Remarks to the Author:

In this paper, authors performed single-cell transcriptome profiling of cancer tissues from patients diagnosed with gastric signet ring cell carcinoma (GSRC). They analyzed 32456 single cells from para-cancerous tissues of 5 patients and 117326 single cells from the cancer tissues of 13 patients. They identified Microseminoprotein Beta (MSMB) as a marker which could guide the signet ring cell carcinoma (SRCC) and moderate/poorly differentiated adenocarcinoma (M/PDA) cells. SRCC were mainly enriched in abnormally active cancer-related signaling pathways and immune response signaling pathways, and in mitogen-activated protein kinases (MAPK) and estrogen signaling pathway. Compared to M/PDA, GSRC has unique cytological characteristics and immune microenvironment. I read this manuscript with great interest. However, the paper suffers from a number of major limitations.

1. The key data quality control steps warrant further check and some of the steps of data processing are potentially problematic. The methods of single-cell data processing is very brief. The doublet removal, evaluation of the potential batch effects, and cell cycle regression were all missing.
2. The authors should provide more information or references about these genes used to annotation since each subgroup, for example, why VWF, CDH5, and PECAM were used to annotate enterochromaffin cells not the endothelial cell?
3. The figure 1 and figure 2 were overlapped and figure 2 were missing some parts, which made confusion to read this paper.
4. The authors used inferCNV analysis to identification of mucous cells and SRCC cells, could the authors explain why the cells having lower degree of variation and CNV score were defined as SRCC cells?
5. Could the authors explain how the MSMB had been found? And why subcluster 7, defined as mucous cells, expressed high level of MSMB, but other mucous subclusters 0, 2, 3, 5, 10, which mainly from the para-cancerous, expressed low level of MSMB? Meanwhile the verification of cell culture, immunohistochemistry, and bioinformatics showed MSMB was higher expression in normal stomach tissue than SRCC cells, which caused dispute with the scRNA-seq finding.
6. The CD38, CD24, CD19 in Dotplot of figure 7A represented which kind of B cells?
7. Follicular B cell were found mainly in GSRC, as well as increased infiltration of CD4-Treg cells and decreased infiltration of CD8-Teff cells, these results lack of validation.
8. The figure captions were too simple to explain the content of figures.
9. The text require revision for thorough correction of typos and grammar, there was no space or more than one space before some words.

Reviewer #2:

Remarks to the Author:

This is a single cell RNA seq study of gastric SRCC cancers. The results are novel and this is a significant contribution to this understudied field.

Several points should be addressed:

1) I am confused how the cohorts (discovery and validation) are set up. The 13 patients - are they all GSRC, or are M/PDA patients part of it to? The figure 1 legend and the text do not explain. There may have been some details in the supplemental section but those files were not made available to the reviewer.

2) In the text of the paper, there is no mention of ethics review prior to study commencement. Did the project have an ethics review?

3) Although there was brief mention of CDH1, can the authors confirm that the GSRC are sporadic, with no known germline mutations in CDH1, alphaE-cadherin, etc?

4) No statements are made with respect to funding, conflict of interest, or data availability. While this might be hidden as part of blinded review, this is critical information.

Responses to the comments

Reviewer #1, expertise in scRNAseq, TME, gastric cancer (Remarks to the Author):

In this paper, authors performed single-cell transcriptome profiling of cancer tissues from patients diagnosed with gastric signet ring cell carcinoma (GSRC). They analyzed 32456 single cells from para-cancerous tissues of 5 patients and 117326 single cells from the cancer tissues of 13 patients. They identified Microseminoprotein Beta (MSMB) as a marker which could guide the signet ring cell carcinoma (SRCC) and moderate/poorly differentiated adenocarcinoma (M/PDA) cells. SRCC were mainly enriched in abnormally active cancer-related signaling pathways and immune response signaling pathways, and in mitogen-activated protein kinases (MAPK) and estrogen signaling pathway. Compared to M/PDA, GSRC has unique cytological characteristics and immune microenvironment. I read this manuscript with great interest. However, the paper suffers from a number of major limitations.

1.The key data quality control steps warrant further check and some of the steps of data processing are potentially problematic. The methods of single-cell data processing is very brief. The doublet removal, evaluation of the potential batch effects, and cell cycle regression were all missing.

Response: In the last version of the manuscript, due to the limitation of the number of words, we have made some deletions. This manuscript is modified as follows and the corresponding key links are verified:

Cell Ranger software (10X Genomics) was used to analyze the sequencing data and gene expression information was obtained for each cell. The Cell Ranger output was loaded onto Seurat 3.1.1 software for dimensionality reduction, clustering, and analysis of scRNA-seq data. Overall, 149,782 cells passed the quality control threshold, in which all genes expressed in less than three cells were removed, the number of genes expressed per cell > 500 was considered as low cut-off and <5000

as high cut-off, with a unique molecular identifier (UMI) count less than 500, and the rate of mitochondrial-DNA derived gene-expression was <25%. DoubletFinder was used to remove multiple cells from sequencing data.

To visualize data, the LogNormalize method of the "Normalization" function of the Seurat software was utilized to calculate the expression levels of genes. PCA was performed using the normalized expression levels, and the top 10 PCs were used to carry out clustering and t-distributed stochastic neighbor embedding (t-SNE) analysis. Due to the obvious batch effect among samples, Harmony was used to correct the batch effect.

We analyzed the impact of cell cycle and found that the overall sample was less affected by cell cycle. From the tSNE diagram, the sample clustering was not significantly affected by the cell cycle. Therefore, we did not remove the impact of cell cycle, so as not to lose the difference between different cell types.

2. The authors should provide more information or references about these genes used to annotation since each subgroup, for example, why VWF, CDH5, and PECAM were used to annotate enterochromaffin cells not the endothelial cell?

Response: We have provided references about these genes used to annotation in the revised manuscript. VWF, CDH5, and PECAM were used to annotate endothelial cells, but not enterochromaffin cells. Enterochromaffin cells were written wrongly because of a clerical error.

3.The figure 1 and figure 2 were overlapped and figure 2 were missing some parts, which made confusion to read this paper.

Response: We apologize for this mistake. Due to our typographical error, images were overlapped, and this mistake was corrected in the revised manuscript.

4. The authors used inferCNV analysis to identification of mucous cells and SRCC cells, could the authors explain why the cells having lower degree of

variation and CNV score were defined as SRCC cells?

Response: We identified mucous and SRCC cells mainly from tissue origin, sub-clusters of 1, 4, 8, and 15 were SRCC cells, and sub-clusters of 0, 2, 3, 5, 7, 10, and 17 were mucous cells (Fig. 1a). In order to verify the correctness of the identification, we conducted cancer-related score (Fig. 1b), and GSVA of inferCNV and CNV score (Fig. 1c).

According to inferCNV analysis, we could distinguish non-malignant epithelial cells from M/PDA cells (Fig. 1d). The CNV scores of sub-clusters of 6, 11, 12, and 13 were relatively high, which were identified as M/PDA cells and were consistent with the typical characteristics of malignant tumors. The CNV scores of each sub-cluster were calculated based on the inferCNV. In addition, CNV score gave corresponding confirmation. This was consistent with previous GA-related studies, and further confirmed the correctness of our verification method. Based on the above-mentioned analysis, we applied the same method to the identification of mucous and SRCC cells. It was revealed that CNV score of SRCC cells was relatively lower than M/PDA and mucous cells, while higher than non-malignant epithelial cells (t-test, $P < 0.001$) (Fig. 1e-f). InferCNV estimated the single cell CNV spectrum to distinguish tumor cells from normal epithelial cells, which is not related to the malignancy of tumors. In general, tumor cells are prone to be CNV mutations, and genes in the regions where CNV changes occur are always characterized with over-expression or down-expression compared with normal cells¹. Previous studies had shown that the level of CNV does not indicate the benign GA and malignant GA cells, which is consistent with another single-cell analysis, in which only 25.0% of static malignant cells exhibited high levels of CNV in gastric cancer².

To sum up, inferCNV and CNV score provide a favorable support for the regrouping of epithelial cells.

Fig. 1 Identification of mucous and SRCC cells. **a** tSNE plots of mucous cells of sub-clusters, according to tissue origin. Sub-clusters of 1, 4, 8, and 15 were independently derived from cancer tissues, which were SRCC cells, and sub-clusters of 0, 2, 3, 5, 7, 10, and 17 were mucous cells. **b** Histogram showing distribution of cancer-related score (average expression levels of cancer-related epithelial marker genes) for cells categorized as mucous cells or SRCC cells (t-test, $P < 2.2 \times 10^{-16}$). **c** GSVA was also performed to characterize the different sources of mucous and SRCC cells. **d** Heat map showing large-scale copy number variation (CNV) of each sub-cluster of epithelial cells (excluding mucous cells): Red: amplification; blue: deletion. CNV occurred in each sub-cluster of epithelial cells, and the most significant CNV was identified as M/PDA cells. **e** CNV score plot of each sub-cluster of epithelial cells. **f** CNV score plot of epithelial cells from all sub-clusters (colored by cell type). CNV score of SRCC cells was relatively lower than M/PDA and mucous cells, while higher than non-malignant epithelial cells (t-test, $P < 2.2 \times 10^{-16}$).

5. Could the authors explain how the MSMB had been found? And why subcluster 7, defined as mucous cells, expressed high level of MSMB, but other mucous subclusters 0, 2, 3, 5, 10, which mainly from the para-cancerous, expressed low level of MSMB? Meanwhile the verification of cell culture, immunohistochemistry, and bioinformatics showed MSMB was higher expression in normal stomach tissue than SRCC cells, which caused dispute with the scRNA-seq finding.

Response: After we determined sub-groups of SRCC cells, we attempted to find the marker gene. Among top 10 genes with the highest expression levels in each subgroup, MSMB was found as a potential marker gene of GSRC (Fig. 3a-b). The expression level of MSMB was basically consistent with that in the sub-clusters of SRCC cells. MSMB had basically no/low expression level in M/PDA cells^{5,6}. The verification was correspondingly carried out by cell lines. The SRCC cell line, NUGC4, exhibited a higher expression level of MSMB than poorly adenocarcinoma cell line MKN-45 (Fig. 4c-d). Verification was conducted using immunohistochemistry (IHC), and the results confirmed that MSMB expression level in SRCC cells was higher than that in M/PDA cells, whereas lower than that in the gastric foveal proliferation area in the para-cancerous tissues (Fig. 4e). In conclusion, these data supported high expression levels of MSMB in SRCC cells.

Combined with relevant studies, SRCC may originate from MUC5AC⁻/low MUC6⁻ pre-pit cells in the proliferative zone of gastric glands³, SRCC of gastric foveolar epithelium is positive for MUC1, MUC5AC, and MUC6, SRCC derived from gastric foveolar epithelium is originated from the proliferative region of the bottom of the gastric pit and gland neck⁴. IHC results confirmed MSMB expression level in the gastric foveal proliferation area. The UMAP plot of MSMB also revealed that sub-clusters 7 and 8 were adjacent, and they were closely associated together. Thus, it was inferred that a high MSMB expression level was detected in sub-cluster 7, while the sub-clusters of 0, 2, 3, 5, 10, and 17 were derived from mucous cells in other parts. SRCC may originate from sub-cluster 7.

ScRNA-seq results showed that MSMB expression level was relatively higher in SRCC cells than mucous and M/PDA cells (Fig. 3f). IHC results confirmed that MSMB expression level in the gastric foveal proliferation area was relatively high, not in all non-malignant epithelial cells.

Therefore, MSMB was found as a potential marker gene of GSRC, which could be related to the differentiation and development of GSRC, and its specific biological function deserves further research.

Fig. 2. The expression level of MSMB in GA and para-cancerous tissues. **A** The expression level of MSMB in sub-clusters. **b** UMAP plot of the expression level of MSMB in sub-clusters. **c** TPM value of transcript of MSMB in MKN-45 and NUGC4 cells by cell culture. **d** TPM value of transcript of MSMB in cell lines by DepMap database. **e** IHC plots of MSMB in GA and para-cancerous tissues: 1: PDA; 2: PDA with SRCC (PDSRCC); 3:GSRC; 4: para-cancerous tissues. **f** MSMB expression level in cancer and para-cancerous tissues by cell type.

6. The CD38, CD24, CD19 in Dotplot of figure 7A represented which kind of B cells?

Response: Naive B cells (expressing CD19, CD24, and CD38) were referred to “Single-cell profiling of infiltrating B cells and tertiary lymphoid structures in the TME of gastric adenocarcinomas”¹. However, Naive B cells mainly exist in PBMCs. We failed to pass the identification, thus, it was not analyzed in the article. We updated correlation marker gene plot of B cells as follows:

Fig. 3 Dotplot showing the expression levels of cell type marker genes in 4 cell types.

Mucosa-associated lymphoid tissue-derived B (MALT-B) cells (expressing Ig and JCHAIN), follicular B cells (expressing MS4A1, CD74, and HLA-DRA), plasma cells (expressing Ig-related genes and IGHG1), and memory B cells (expressing HMGN2 and H2AFZ).

7. Follicular B cell were found mainly in GSRC, as well as increased infiltration of CD4-Treg cells and decreased infiltration of CD8-Teff cells, these results lack of validation.

Response: Follicular B cells were mainly found in GSRC, as well as increased infiltration of CD4-Treg cells and decreased infiltration of CD8-Teff cells. We carried

out relevant verification by GSRC (n=30) and M/PDA (n=30) samples, which were used for immunohistochemical verification. We used the main marker genes of each type of cells to verify: follicular B cells (expressing CD74), MALT-B cells (expressing JCHAIN), CD4-Treg cells (expressing FOXP3), and CD8-Teff cells (expressing KLRD1) (Fig. 5).

Fig. 4 Differential plot of infiltration of follicular B and MALT-B cells/CD4-Treg and CD8-Teff cells in M/PDA and GSRC by immunohistochemistry score. a Compared with M/PDA, the immunohistochemical score of follicular B cells (expressing CD74) in GSRC was higher, and the difference was statistically significant (Wilcoxon rank-sum test, $P=0.012$); The immunohistochemical score of MALT B cells (expressing JCHAIN) in GSRC was higher, and the difference was not statistically significant (Wilcoxon rank-sum test, $P=0.154$). **b** Compared with M/PDA, the immunohistochemical score of CD4-Treg cells (expressing FOXP3) in GSRC was higher, and the difference was statistically significant (Wilcoxon rank-sum test, $P=0.032$); The immunohistochemical score of CD8-Teff cells (expressing KLRD1) in GSRC was lower, and the difference was statistically significant (Wilcoxon rank-sum test, $P=0.021$).

8. The figure captions were too simple to explain the content of figures.

Response: We have re-written this part according to your comment as follows:

Fig. 1 ScRNA-seq overview and identification of major cell types in GA and

para-cancerous tissues. a Schematic diagram of scRNA-seq process. **b** Experimental design for scRNA-seq and corresponding validation. In the discovery cohort, cancer tissues (n=13) and para-cancerous tissues (n=5) of 13 patients with GA who underwent radical gastrectomy were used for analysis. In the validation cohort, GSRC (n=30) and M/PDA (n=30) samples were used for IHC verification. The mRNA transcriptome was sequenced by cell culture. Relevant databases

were used for bioinformatics validation. **c** The t-distributed stochastic neighbor embedding (t-SNE) plot of the 26 identified main cell types from cancer and para-cancerous tissues. Among them, 32,456 single cells were obtained from para-cancerous tissues, and 117,326 single cells were obtained from cancer tissues. **d** Violin diagram shows the expression levels of cell type maker genes in 8 cell types. **e** tSNE plot of cells from cancer and para-cancerous tissues (colored by cell type). **f** tSNE plot of cells from cancer and para-cancerous tissues (colored by sample origin). **g** tSNE plot of cells from cancer tissues, which were grouped by the degree of differentiation and the content of signet ring cells. **h** Scale plot of cells from cancer tissues, which were grouped by the degree of differentiation and the content of signet ring cells.

Fig. 2

9. The text require revision for thorough correction of typos and grammar, there was no space or more than one space before some words.

Response: Thanks for your comment. The manuscript was fully edited.

Reviewer #2, clinical expertise in gastric cancer (Remarks to the Author):

This is a single-cell RNA-seq study of SRCC. The results are novel and this is a significant contribution to this understudied field.

Several points should be addressed:

1. I am confused how the cohorts (discovery and validation) are set up. The 13 patients - are they all GSRC, or are M/PDA patients part of it to? The figure 1 legend and the text do not explain. There may have been some details in the supplemental section but those files were not made available to the reviewer.

Response: There have been some details in the supplemental section, and we have added corresponding contents to the legend of Figure 1.

2. In the text of the paper, there is no mention of ethics review prior to study commencement. Did the project have an ethics review?

Response: Thanks for your comment. The present study was approved by the Institutional Review Boards of Shandong Cancer Institute (Approval No.

SDTHEC2020011006) and Jinan People's Hospital (Approval No. GZR2019-041-01). All specimens were collected from patients who gave written informed consent. This research was conducted according to the principles of the Declaration of Helsinki.

3. Although there was brief mention of CDH1, can the authors confirm that the GSRC are sporadic, with no known germline mutations in CDH1, alphaE-cadherin, etc?

Response: Studies have found that the gene change of CDH1 is closely related to familial gastric cancer, and is closely related to the occurrence and evolution of GSRC^{7,8}. According clinicopathological characteristics of GSRC specimen, all cases in our study may occur sporadically with no family history of the condition. However, we found that CDH1 expression level was basically the same between SRCC cells and adenocarcinoma cells. One explanation could be the frequency of CDH1 mutations increased in GSRC⁹. Furthermore, our study found that the immunoglobulin superfamily was crucial in the cell adhesion of SRCC cells. Compared with adenocarcinoma cells, the intercellular adhesion function decreased in SRCC cells. However, the intercellular adhesion function decreased or destroyed, which is the key in causing cancer cells to break away from the parent tumor to initiate metastasis and may also be associated with higher risks of invasive growth and metastasis, especially for implanted metastasis of GSRC.

4. No statements are made with respect to funding, conflict of interest, or data availability. While this might be hidden as part of blinded review, this is critical information.

Response: We had corresponding data in supplementary materials (This might be hidden as part of blinded review), and uploaded according to relevant regulations.

References

1. Jia, L. et al. Single-cell profiling of infiltrating B cells and tertiary lymphoid structures in the TME of gastric adenocarcinomas. *Oncoimmunology* **10**, 1969767 (2021).
2. Zhang, M. et al. Dissecting transcriptional heterogeneity in primary gastric adenocarcinoma by

- single cell RNA sequencing. *Gut* **70**, 464-475 (2021).
3. Zhou, Z. H., Zhang, J. D., Zhao, H. B., Zhao, L. N. & Shan, B. Z. Cell origin and premalignant lesions of gastric signet-ring cell carcinoma: A histopathologic study. *World Chinese Journal of Digestology* **18**, 2001-2006 (2010).
 4. Zhang, Z. S. et al. Clinicopathological characteristics of signet-ring cell carcinoma derived from gastric foveolar epithelium. *J Dig Dis.* **23**, 396-403 (2022).
 5. Ohnuma, S. et al. Cancer-associated splicing variants of the CDCA1 and MSMB genes expressed in cancer cell lines and surgically resected gastric cancer tissues. *Surgery* **145**, 57-68 (2009).
 6. Meng, L. et al. Identification of gastric cancer-related genes by multiple high throughput analysis and data mining. *Zhonghua Wei Chang Wai Ke Za Zhi* **10**, 169-172 (2007).
 7. Muta, H. et al. E-cadherin gene mutations in signet ring cell carcinoma of the stomach. *Japanese journal of cancer research* **87**, 843-848 (1996).
 8. Chen, J. et al. Screening of differential microRNA expression in gastric signet ring cell and gastric adenocarcinoma and target gene prediction. *Oncology reports* **33**, 2963-2971 (2015).
 9. Humar, B. et al. E-cadherin deficiency initiates gastric signet-ring cell carcinoma in mice and man. *Cancer Res.* **69**, 2050-2056 (2009).

Reviewers' Comments:

Reviewer #1:

Remarks to the Author:

In this revised manuscript, the authors involved doublet removal, evaluation of the potential batch effects, and cell cycle regression in methods, re-performed the inferCNV analysis to mucous and SRCC cells, explained and verified MSMB as a potential marker gene of GSRC, and validated follicular B cells were mainly found in GSRC, as well as increased infiltration of CD4-Treg cells and decreased infiltration of CD8-Teff cells. They have solved most of my doubts, but there remained a few problems needed to be discussed.

1. The authors responded that they had provided references about marker genes used to annotation in the revised manuscript, but there had no references founded in the revised manuscript.
2. The IHC validation of MSMB in Figure 4I lacks a scale ruler, and four of the IHC results seems were not presented in the same scale.

Reviewer #2:

Remarks to the Author:

Thank you for addressing these comments. I look forward to seeing the paper published.

Recommend accepting for publication.

Responses to the comments

Reviewer #1 (Remarks to the Author):

We thank you for the thoughtful review and comments on our study. We have fully revised our manuscript and provided responses to each comment in detailed below.

1. The authors responded that they had provided references about marker genes used to annotation in the revised manuscript, but there had no references founded in the revised manuscript.

Response: we had provided references about marker genes used to annotation in the last version of the manuscript. Four references have been added, eg : They were named according to specific marker genes: B cells (expressing *CD79B*, *CD79A*, and *MS4A1*), endothelial cells (expressing *VWF*, *CDH5*, and *PECAM*), epithelial cells (expressing *CDH1*, *KRT8*, and *EPCAM*), fibroblast cells (expressing *PDGFRB*, *COL1O2*, and *DCN*), mast cells (expressing *SLC18A2*, *FCER1A*, *TPSB2*, and *KIT*), myoid cells (expressing *FCGR2A*, *CD163*, and *MRC1*), smooth muscle cells (expressing *TAGLN*, *RGS5*, and *ACTA2*) and T cells (expressing *TRBC2*, *CD2*, and *CD3E*)¹⁹, Through cluster analysis of epithelial cells, 20 sub-clusters of epithelial cells were obtained. According to marker genes cancer cells (expressing *CLDN4*, *REG4*, *TTF3*, and *CEACAM6*), mucous cells (expressing *TFF1*, *MUC5AC*, *TFF2*, and *MUC6*), chief cells (expressing *PGA3* and *PGA4*), parietal cells (expressing *ATP4A* and *ATP4B*), and endocrine cells (expressing *CHGA* and *CHGB*)¹⁹,etc.

2. The IHC validation of MSMB in Figure 4I lacks a scale ruler, and four of the IHC results seems were not presented in the same scale.

Response: In Figure 4I, the figure of 1, 2, and 3: Scale bar: 50µm. The figure of 4: Scale bar: 5µm.

Thank you very much for your great supports.

Jie Chai, MD, PhD